# LANDSLIDE SUSCEPTIBILITY MAPPING BY USING GIS ALONG THE CHINA PAKISTAN ECONOMIC CORRIDOR (KARAKORAM HIGHWAY), PAKISTAN.

Sajid Ali[1-2], Peter Biermanns[1], Rashid Haider[3] and Klaus Reicherter[1]

1Neotectonics and Natural Hazards, RWTH Aachen University, Lochnerstr. 4-20, 52056 Aachen, Germany.
2Department of Earth Sciences, COMSATS Information Technology, Abbottabad, Pakistan.
3Geological Survey of Pakistan, Islamabad, Pakistan.

*Correspondence to*: Sajid Ali (s.ali@nug.rwth-aachen.de)

**Abstract.** The Karakoram Highway (KKH) is an important route, which connects Northern Pakistan with Western China. Presence of steep slopes, active faults and seismic zones, sheared rockmass and torrential rainfalls make the study area a unique geohazards laboratory. Since its construction, landslides constitute an appreciable threat, having blocked the KKH for several times. Therefore, landslide susceptibility mapping was carried out in this study, to support Highway authorities in maintaining smooth and hazard free travelling. Geological and geomorphological data were collected and processed using GIS environment. Different conditioning and triggering factors for landslide occurrences were considered for preparation of the susceptibility map. These factors include lithology, seismicity, rainfall intensity, faults, elevation, slope angle, aspect, curvature, land cover and hydrology. According to spatial and statistical analyses, active faults, seismicity and slope angle mainly control the spatial distribution of landslides. Each controlling parameter was assigned a numerical weight by utilizing the Analytic Hierarchy Process (AHP) method. Additionally, the weighted overlay method (WOL) was employed to determine landslide susceptibility indices. As a result, the landslide susceptibility map was produced. In the map, the KKH was subdivided into four different susceptibility zones. Some sections of the Highway fall into high to very high susceptibility zones. According to results, active faults, slope gradient, seismicity and lithology have a strong influence on landslide events. Credibility of the map was validated by landslide density analysis (LDA) and receiver operator characteristics (ROC), yielding a predictive accuracy of 72% which is rated as satisfactory by previous researchers.

## 1   Introduction

Landslides are a result of different geodynamic processes and represent a momentous type of geohazard, causing economic and social loss by damaging infrastructure and buildings (Vallejo and Ferrer, 2011). Landslides are mainly caused by conditioning and triggering factors. Conditioning factors include relief, lithology, geological structure, geomechanical properties and weathering, whereas precipitation, seismicity, change in temperature and static or dynamic loads are triggering factors. Variations in these factors affect the occurrence of landslides. Heterogeneity in lithology influences hydrological and mechanical characteristics of rockmass. Slope morphology (curvature) depends upon lithology and structure within it. Size and type of mass movement changes with variations in lithology and structures. Some lithologies are more permeable and

allow water to infiltrate and to increase the pore water pressure. This increase in pore water pressure ultimately affects shear strength of the rockmass and slope stability (Barchi et al., 1993; Cardinali et al., 1994). Whereas, during rainfall event, less pervious rockmass has low infiltration and high runoff leading to debris and mud flows (Dramis et al., 1988b; Ellen, 1988, Canuti, 1993). Sheared and highly jointed rockmass contains shallow slope failures whereas rockfalls are concentrated in well-
bedded massive rockmass (Jin et al., 1991; Hu and Cruden, 1993). Distance from a tectonic feature has an inverse relation with rock fracturing and degree of weathering (Pradhan et al., 2010). State of weathering and fracturing makes slopes unstable (Ruff and Czurda, 2008). Slope is an important driving parameter for slope failures in same geological and climatic setting (Coco and Buccolini, 2015). Shear strength decreases with increase in slope. Therefore, landslide density increases with increase in steepness (Pradhan et al., 2010). Symposium et al., (2006) found more than half of landslides in areas where angle
is greater than 25%. Curvature value expresses the shape of the slope. If it is positive then slope will be upwardly convex and will be concave in case of negative value. Later has ability to retain the water for longer time leading to increase pore water pressure and hence into slope failures (Pradhan et al., 2010). Assessment of risk related to landslides remained a long-time challenge for geologists but was significantly facilitated with the eventual availability of remote sensing data (Shahabi and Hashim, 2015). Preparation of landslide inventory maps, acquisition of geomorphological data (elevation, slope, slope
curvatures, aspect), hydrological parameters and extraction of lineaments from remote sensing products is now comparatively an easier task.

Landslide susceptibility mapping is the spatial prediction of landslide occurrence by considering causes of previous events (Guzzetti et al., 1999). It largely depends upon the knowledge of slope movement and controlling factors (Yalcin, 2008). It has hitherto been carried out by many researchers in order to denominate potential landslide hazard zones through evaluation
of responsible factors (Basharat et al., 2016; Komac, 2006; Lee et al., 2002, 2004; Shahabi and Hashim, 2015; Süzen and Doyuran, 2004). Preparation of their maps was based broadly on qualitative and quantitative approaches. Early research work (Nilsen et al., 1979) was largely quantitative, utilizing deterministic and statistical correlations and regression analysis of landslides and their controlling factors. In this context, safety factors, calculated on the base of engineering parameters are adduced to imply deterministic methods. In more recent works, statistical methods are favoured, attempting correlation
between spatial distribution of landslides and their controlling factors. Among these are e.g. analytical hierarchy process, bivariate, multi variate, logistic regression neural network, fuzzy logic etc. (Basharat et al., 2016; Guzzetti et al., 1999; Komac, 2006; Lee et al., 2002, 2004; Shahabi and Hashim, 2015; Süzen and Doyuran, 2004). These techniques were proven to be better options for comparatively large and complex areas (Cardinali et al., 2000). Expert opinion and landslide inventories are the decisive components of qualitative approaches (Yalcin, 2008). In most cases, landslide inventories were adduced to
estimate failure susceptibility based on previous hazards in locations with similar geological, geomorphological and hydrological setups. Some geoscientists (Ahmed et al., 2014; Ayalew et al., 2004; Basharat et al., 2016; Kamp et al., 2008; Kanwal et al., 2016; Shahabi and Hashim, 2015; Yalcin, 2008) incorporated statistical techniques (Analytical Hierarchy Approach (AHP) with Weighted Linear Combination (WLC) and Weighted Overlay Method (WOM)) in qualitative methods

to provide the identified factors with a numerical weightage. The combination of AHP and Weighted Overlay Method (WOM) was termed as Multi Criteria Decision Analysis (MCDA) (Ahmed, 2015; Basharat et al., 2016; Kanwal et al., 2016). AHP is a simple and flexible method to analyze and solve complex problems (Saaty, 1987, 1990). It facilitates the estimation of influence that different factors might have on landslide development by comparing them in possible pairs in a matrix. This

approach involves field experience and knowledge background of the researcher. Field campaign along the KKH in May 2016 enhanced our knowledge about factors controlling landslides events. Previous research considered MCDA a better choice because of its accuracy to predict landslide hazard (Ahmed, 2015; Basharat et al., 2016; Kamp et al., 2008; Komac, 2006; Park et al., 2013; Pourghasemi et al., 2012). MCDA (combination of AHP with qualitative approaches) was declared a better option for regional studies (Soeters and van Westen, 1996). Previous wide usage in landslide susceptibility mappings, high accuracy,

simple process and flexibility according to local variation in landslide controlling parameters compelled us to choose this model. Furthermore, the use of Geographic Information Systems (GIS) facilitated the extraction of geomorphic and hydrological parameters required for susceptibility assessment. Digital Elevation Models (DEM) are commonly processed in GIS to extract crucial parameters for susceptibility assessment such as elevation, slope, aspect, curvature, watershed etc. (Ayalew et al., 2004, 2005; Basharat et al., 2016; Ohlmacher and Davis, 2003; Rozos et al., 2011; Shahabi and Hashim, 2015;

Süzen and Doyuran, 2004). Previously, landslide susceptibility maps of different fragments of Northern Pakistan were prepared (Bacha et al., 2018; Basharat et al., 2016; Ahmed et al., 2014; Kamp et al., 2008; Kanwal et al., 2016; Khan et al., 2018; Rahim et al., 2018) (Table 1). All of them (Basharat et al., 2016; Kamp et al., 2008; Kanwal et al., 2016; Khan et al., 2018; Rahim et al., 2018) used single model based method except two (Ahmed et al., 2014; Bacha et al., 2018) who compared performances of two models. Ahmed et al. (2014), Kanwal et al. (2016) and Rahim et al. (2018) used regional geological map (1:500000) to

produced landslide susceptibility map of the upper Indus basin. Whereas, inventories were based on published rock avalanches map (Kanwal et al., 2016), geomorphological mapping (Ahmed et al., 2014), co-seismic landslides (Basharat et al., 2016) and remote sensing along with filed mapping (Bacha et al., 2018; Khan et al., 2018). Geological, geomorphological and human induced parameters were considered for production of susceptibility map (Table 1).

**Table 1: Previous work in some parts of the Northern Pakistan.**

| Authors | Method | Causative Factors | Study Area |
|---|---|---|---|
| Kamp et al., 2008 | MCE and AHP | Aspect, Elevation, Faults, Lithology Land Cover, Rivers, Roads, Slope, Tributaries, Aspect | District Muzaffarabad |
| Farooq Ahmed et al., 2014 | WOM and Fuzzy logic | Relief Slope, Curvature, Aspect, Rain, Seismic Hazard Faults, Drainage, Ndvi, Geology | Upper Indus watershed |
| Basharat et al., 2016 | MCE and AHP | Aspect, Elevation, Faults, Lithology, Land Cover, Hydrology, Roads, Slope, Curvature | Tehsil Balakot |
| Kanwal et al., 2016 | AHP based heuristic approach | Slope, Aspect, Lithology, Land Cover, Faults, Road Network, Streams | Shiger and Shyok Basin in Karakorum range |
| Khan et al., 2018 | FR | Slope, Aspect, Curvature, Lithology, Land Cover, Faults, Road Network, Distance from Stream, SPI, TWI | Haramosh valley, Bagrote valley and parts of Nagar valley. |
| Bacha et al., 2018 | WOT, FR, | Aspect, Fault, Geology, Land Cover, Proximity to Road, Slope, Proximity to Stream | Hunza-Nagar valley |
| Rahim et al., 2018 | AHP and WLC | Slope, Aspect, Elevation, Drainage Network, SPI, TWI, Lithology, Fault Lines, Rainfall, Road Network, Land Cover, Soil Texture | District Ghizer |
| This Study | AHP and WOM | Elevation, Slope Angle, Aspect, Curvature, Lithology, Seismicity, Faults, Land Cover, Rainfall Intensity And Distance From Streams | KKH (CPEC) |

## 2 General situation of the study area

The Karakoram Highway (KKH), a part of the China-Pakistan Economic Corridor (CPEC), connects Northern Pakistan with Western China (Fig. 1). It passes through rapidly rising mountain ranges of Himalaya, Karakoram, Hindu Kush forming the

5 junction between the Indian and Eurasian plates including Kohistan Island Arc (Derbyshire et al., 2001). The area is characterized by fractured and weathered rockmass, diverse lithologies (igneous, metamorphic, and sedimentary), high seismicity, deep gorges, high relief, arid to Monsoon climate and locally high rates of tectonic activity. These conditions make the study area a unique geohazards laboratory. Starting with its construction in 1979, KKH's stability has been endangered by a variety of geohazards.

10 The study area is the 840 km long (10 km buffer) Highway (KKH), N35, located in the Karakoram Mountains, Himalaya. The area hosts some of the highest reliefs and highest peaks (Nanga Parbat: 8126 m, Rakaposhi: 7788 m) in the world (Hewitt, 1998). Goudie et al. (1984) termed the study area the steepest place on the earth where elevation drops from 7788 m to 2000 m over a horizontal distance of 10 km (Fig. 2b, 2d).

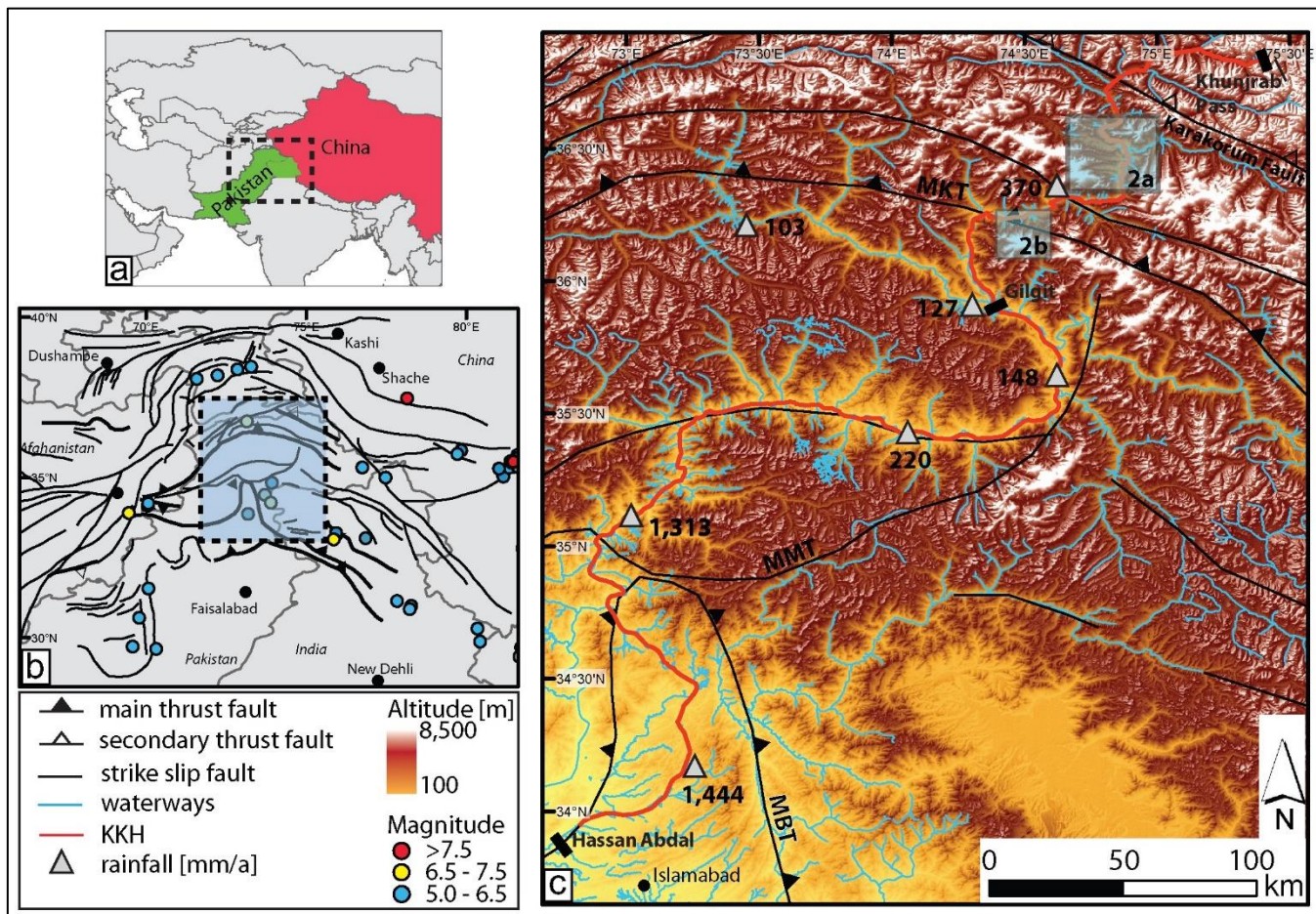

**Figure 1: Overview of tectonics & precipitation in the study area: (a) Location of the study area in the region (b) Active faults and major earthquake events in the region (USGS Earthquake Catalog, 2017) (c) Locations of the weather stations with mean annual rainfall (Pakistan Meteorological Department) and overview of tectonics and topography of the study area (After Hodges 2000): Box 2a and 2b represents location of Figure 2. KKH=Karakoram Highway, MBT=Main Boundary Thrust, MMT=Main Mantle Thrust, MKT=Main Karakoram Thrust (modified after Ali et al., 2017).**

From Abbotabad, the Highway leads northwards through the Sub-Himalayas entering the Indus valley at Thakot, and the Hunza valley at Gilgit, running parallel to the eponymous rivers. From Thakot onwards, it passes through deeply incised valleys and gorges.

Weather conditions along the KKH are not uniform and are characterized by a wide range of annual mean temperatures (-5 °C to 46 °C) and precipitation (15 mm to 1500 mm). The distribution of precipitation is additionally strongly fluctuating throughout the year. During the westerlies (January, February and March) and the monsoon period (July, August), the study area receives heavy rainfall. According to meteorological data, average annual precipitation between Abbottabad and Dassu is 1444 mm. However, north of Dassu, an abrupt change from monsoon to semi-arid to arid conditions is recorded which is owed to a change in valley orientation from north-south to east-west (Fig.1 and Fig. 3). Furthermore, vertical climatic zonation

exists in the Hunza valley along the KKH. The surrounding peaks and slopes higher than 5000 m receive precipitation greater than 1000 mm, whereas the valley floor below is characterized by a semi-arid to arid climate (Hewitt, 1998).

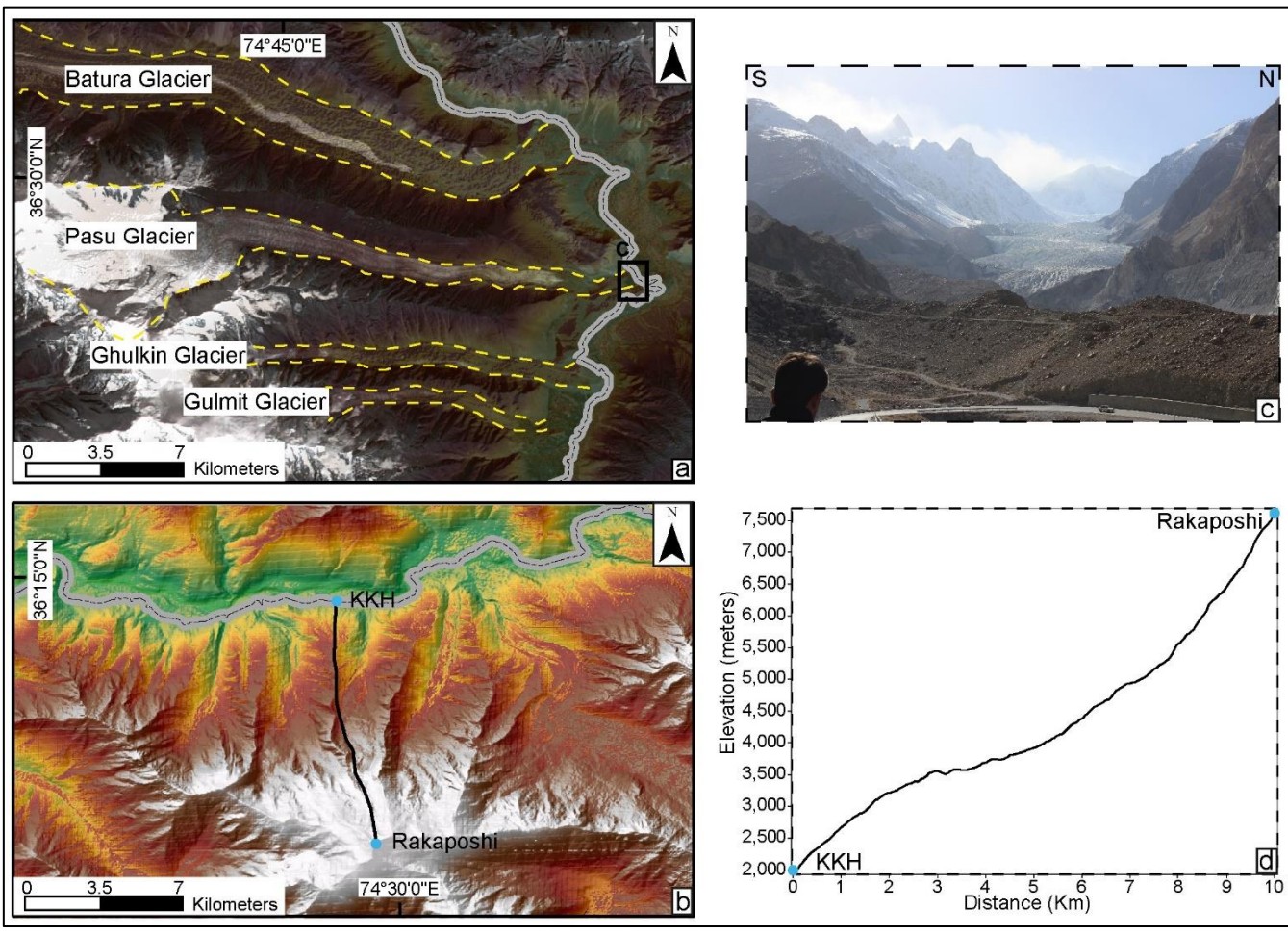

**Figure 2: a) Glaciers along the KKH (b) Relief along the KKH (c) Pasu Glacier's snout approaching the KKH (d) Profile drawn along axis drawn in 2b: elevation drops from 7788 m to 2000 m over a horizontal distance of 10 km.**

The Karakoram and Himalaya host some of the world's longest continental glaciers with the steepest gradient and highest glacial erosion rates (Goudie et al., 1984). The snouts of some glaciers (Batura, Ghulkin, Pasu, Gulmit, Gulkin) are close to the KKH and partly cross it (Fig. 2a, 2c). Relatively warm temperature in the valleys results in sudden melting of ice, frequent surges, catastrophic debris flows and blockage of rivers.

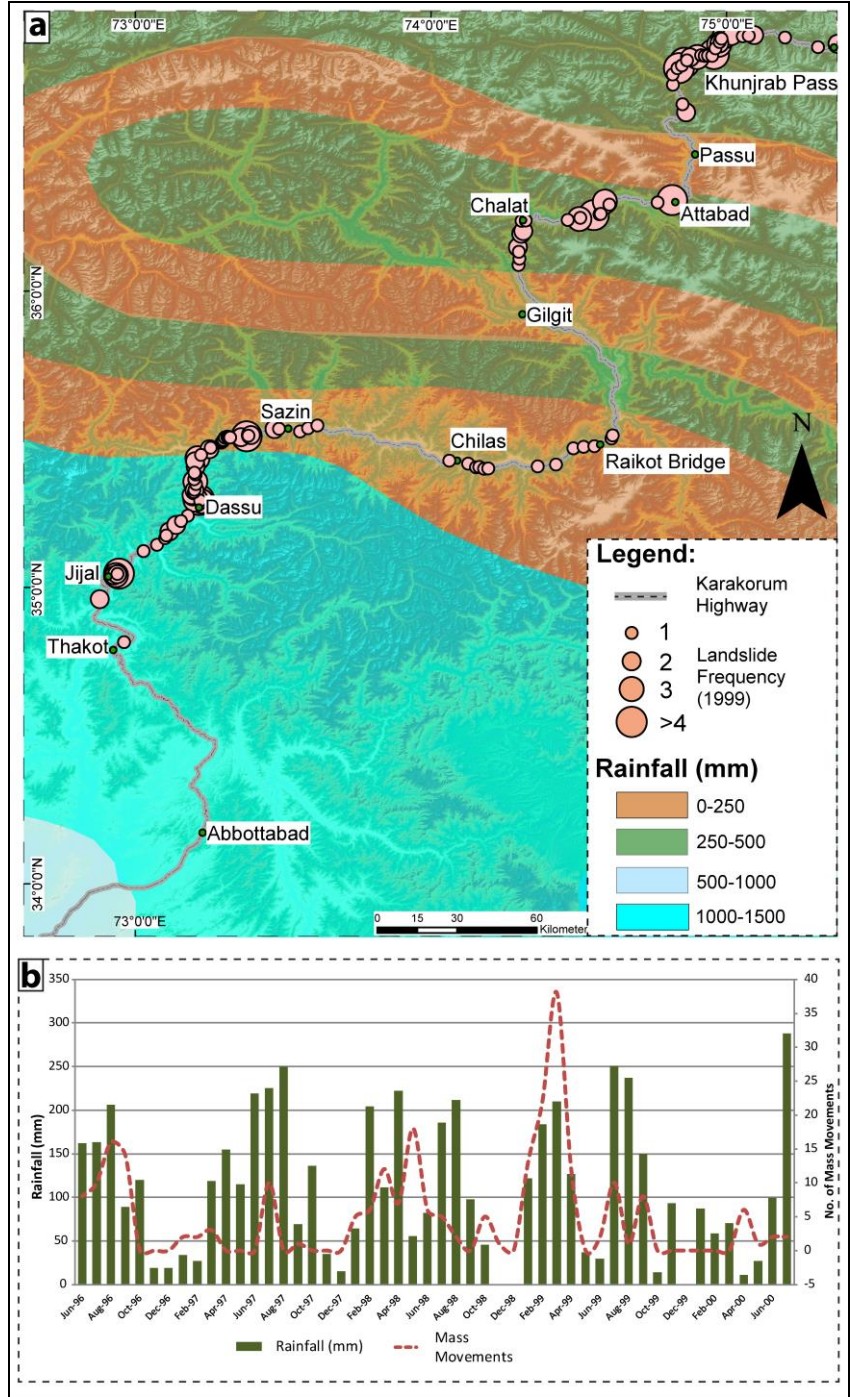

**Figure 3: (a) Overview of precipitation (mean annual rainfall) and landslides frequency along the KKH (after Khan et al., 2000, Pakistan Materological Department 1982, 83, 96, 97, 98, 99, 2000, 14, 15, 16, Frontier Works Organisation archives) (b) Correlation between landslide events and precipitation (Ali et al., 2017).**

## 3    Geology along the KKH

Tectonically, the area is characterized by orogenic features that started forming with the onset of the Indo-Eurasian collision 50 Myr ago. Crustal shortening, subduction, and active faulting are still ongoing with convergence rates of ~4-5 cm/y (Jade, 2004) and uplift rates of ~7mm/y (Zeitler, 1985). Main Mantle Thrust (MMT), Kamila Jal Shear zone (KJSZ), Raikot Fault, Main Karakoram Thrust (MKT) and Karakorum Fault are important tectonic features responsible for brittle deformation along the Highway (Fig. 4) (Bishop et al., 2002; Burg et al., 2006; DiPietro et al., 2000; Goudie et al., 1984). Due to this brittle deformation, the rockmass is highly jointed and fractured. The general geology along the KKH consists of sedimentary, igneous and metamorphic rocks. Highly active landslide zones were identified from the multi-temporal landslide inventory of the KKH (Fig. 6). Jijal-Dassu, Raikot Bridge, Hunza valley and Khunjrab valley sections is characterised by a large number of mass movements and therefore, detailed geology is only discussed for these sections.

The geology of the Jijal-Dassu section is composed of ultramafic and low to high-grade metamorphic rocks. The Mansehra granite, the Besham group, the Jijal complex, the Kamila amphibolite and the Chilas complex are important lithological units in this section. The Besham group comprises biotitic gneisses, cataclastic gneisses and quartzite which were metamorphosed during the Himalayan orogeny ∼65 Ma ago (Ding et al., 2016). It shares a faulted contact with the Jijal complex (Kohistan Island arc) along the northward dipping MMT (Williams, 1989). The ultramafic rocks with garnet granulites and Alpine type metamorphic rocks between Jijal and Pattan are collectively termed as Jijal complex (Tahirkheli and Jan, 1979). The Besham group shows signs of crushing of individual minerals and staining of quartz whereas the Jijal complex is massive and sheared (Khan et al., 1987). Owed to the contacts of the Jijal Complex with the MMT in the north and the Pattan Fault in the south, it is highly tectonised and deformed. The Kamila amphibolite consists of sheared basic lavas and intrusive plutons (Treloar et al., 1996). It is classified into two types: garnet bearing and garnet free amphibolites. The former is massive and sheared due to the presence of Pattan and Kamila Jal shear zones (KJS), whereas the latter is banded. The garnet free amphibolite shares a sheared contact with the Chilas complex, mafic intrusions of predominantly gabbro-norites, sheared gabbro-norites and diorites (Searle et al., 1999).

The Raikot Bridge section exhibits continuous mass movement process because of its location at the seismically active western limb of the MMT known as Raikot fault, a strike-slip fault with right lateral movement. It is marked by concentration of hot springs and a large shear zone.  Granitic gneisses, quartzites, gabronorite, schists and Quaternary sediments are the main lithological units in this section. Continuous erosion by Indus River and highly deformed rocks are responsible for landslide events.

MKT is a part of the Hunza valley section and is responsible for many landslide events. The pre-dominant local lithologies of the Baltit Group, Chalt schists, Karakoram batholith as well as Quaternary sediments are highly tectonised and deformed.

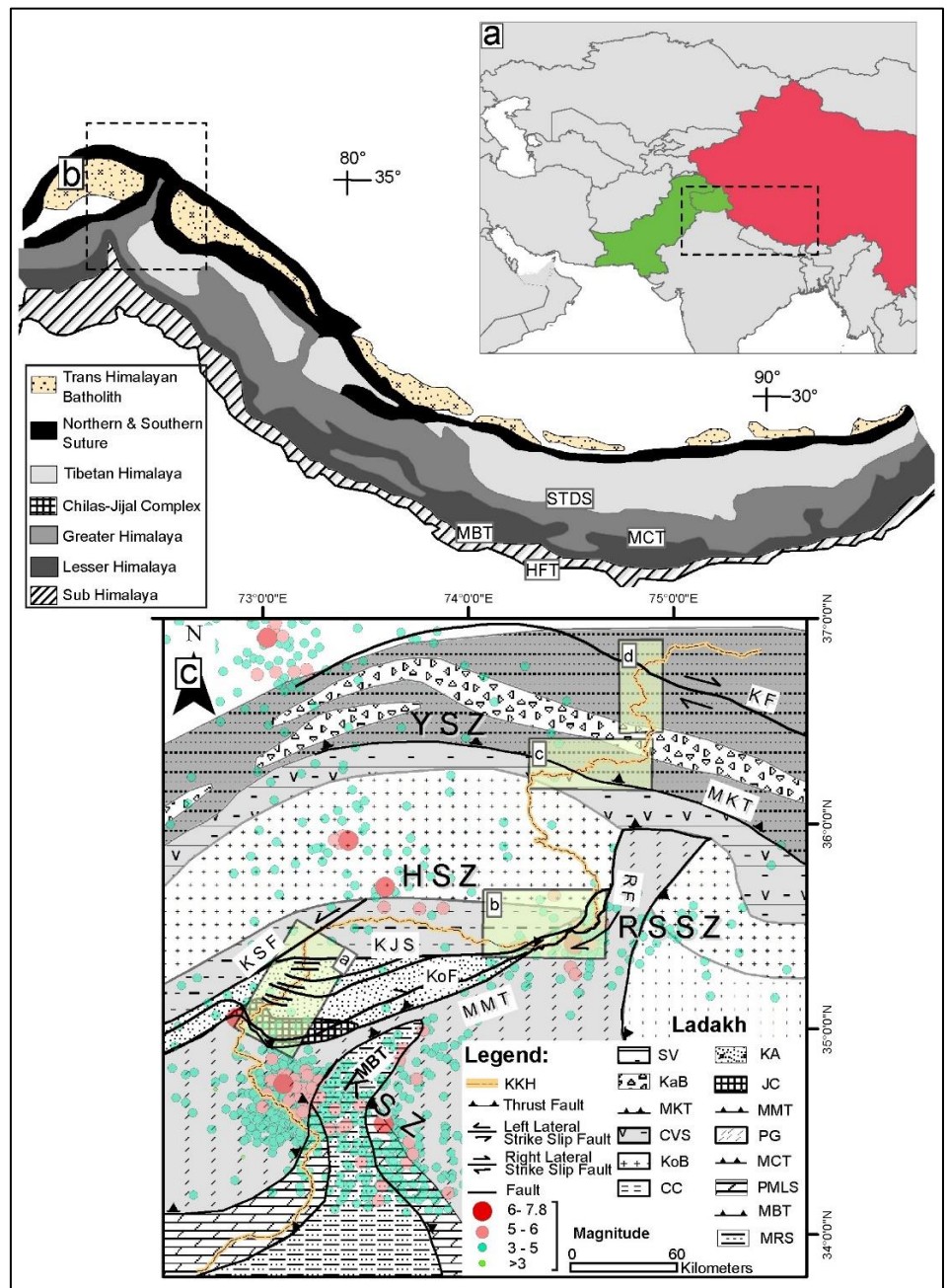

**Figure 4: (a) Regional location of Himalaya (b) Overview of Himalayan geology (c) Geology along the KKH (compiled from: Khan and Jan, 1991; Derbyshire et al., 2001; DiPietro and Pogue, 2004; DiPietro et al., 2000; Hewitt et al., 2011). Four boxes represent four sections (a-Jijal- Dassu Section, b-Raikot bridge Section, c- Hunza Valley Section, d-Khunjrab valley Section) MMT-Main Mantle Thrust, KJS-Kamila Jal Shear zone, MKT- Main Karakoram Thrust, KF- Karakoram Fault, KSF-Kamila Strike slip fault, IKSZ-Indus Kohistan Seismic zone, HSZ- Harman Seismic Zone, RSSZ-Raikot Sassi Seismic zone, YSZ-Yasin Seismic Zone, SV-Sediments and Volcanics, KaB-Karakoram Batholith, CVS-Chalt Volcanics and Schist, KoB-Kohistan Batholith, CC-Chilas Complex, KA-Kamila Amphibolites, JC, Jijal Complex, PG-Precambrian Gneisses, PMLS-Palaeozoic and Mesozoic Limestones and Sandstones, MRS-Miocene Redstones.**

The highly deformed Misgar slates, along with the Gujhal dolomite and Kilk formation, are the main components of the Sost section. The Karakoram fault is an important tectonic feature in this section. Highly fissile and closely jointed slates are important sources of scree on steep slopes along the Highway. Intense weather conditions aggravate the situation in this section.

## 4    Seismology

The Highway passes through one of the seismically most active areas in the world. The presence of active thrusts and strike slip faults gives rise to earthquakes, anon triggering numerous landslides. The seismic activity along KKH is demonstrated by 317 M>5 and 10 M>7 recorded earthquake events (Muzaffarabad Oct, 2005: M=7.6, Afghanistan Oct, 2015: M=7.5) since 1904 (Zhiquan and Yingyan, 2016). The Highway passes through important seismic zones: the Indus Kohistan seismic zone (IKSZ), the Hamran seismic zone (HSZ), the Raikot-Sassi seismic zone (RSSZ) and the Yasin seismic zone (YSZ) (Fig. 4).

The Jijal-Dassu section of the KKH passes through the northern part of IKSZ. IKSZ is 50 km wide and represents a highly active wedge-shaped structure containing a shallow and midcrustal zone (MonaLisa et al., 2009). The Muzaffarabad (2005, M=7.6) and Pattan earthquakes (1974, M=6.2) are recent destructive earthquakes in this seismic zone.

Sazin section of the Highway is a part of HSZ, an active seismic zone hosting recent events with magnitudes of 3 to 6.2. The active Raikot fault traverses RSSZ and is responsible for shallow seismicity of magnitudes 3 to 6.3. Both the fault and the

KKH run, in direct vicinity, on the western banks of Indus River. The YSZ encompasses the region surrounding the small town of Yasin. It is characterized by earthquakes with magnitudes between 3 and 5 and focal depths of less than 50 km. The MKT is suspected to be the main source of seismicity for this seismically active region.

## 5    Methodology

The flow chart (Fig. 5) describes the steps and techniques involved in preparation of the susceptibility map, involving multiple

techniques, literature review, field observation and remote sensing.

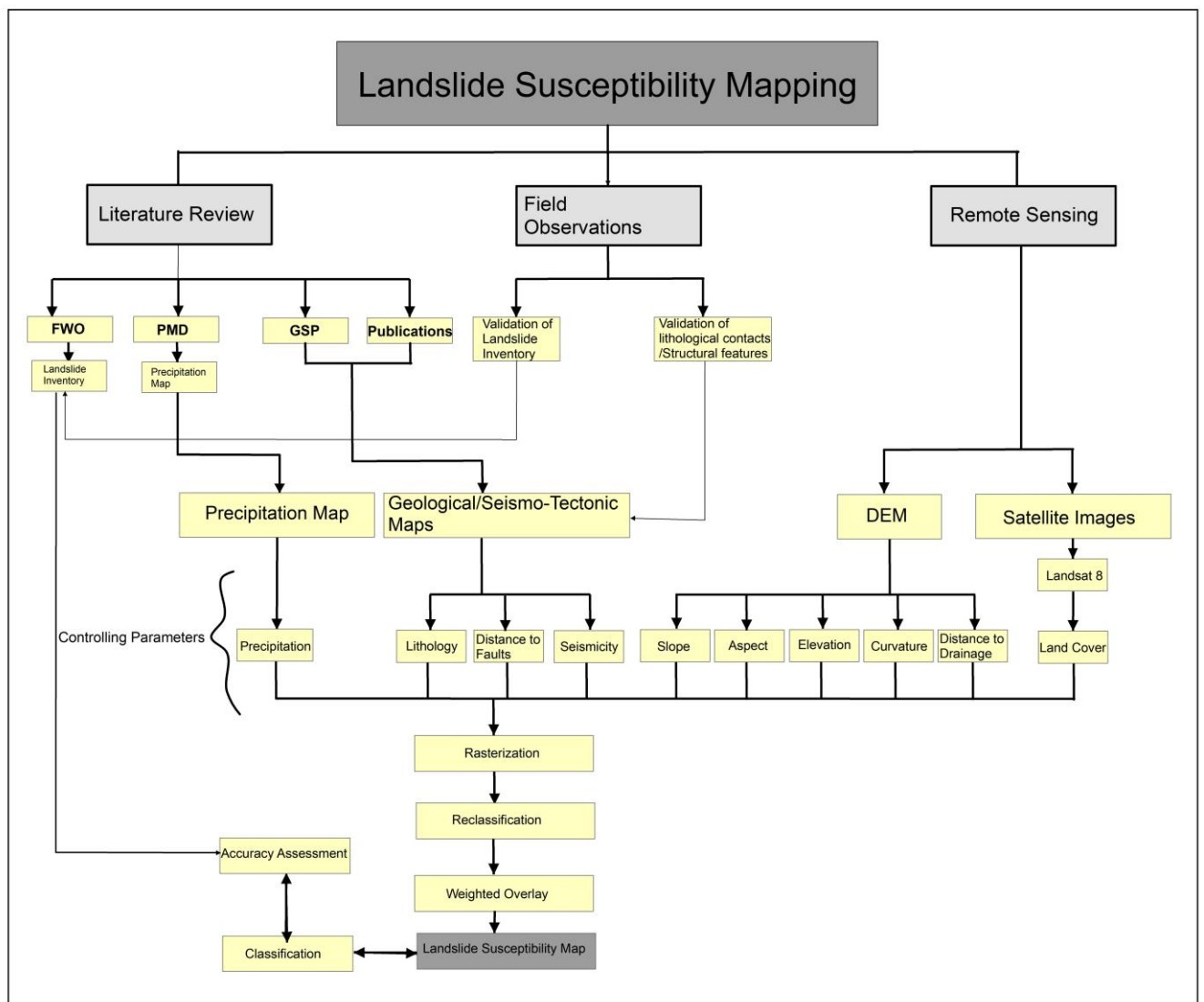

**Figure 5: Flow Chart showing multiple steps involved in the preparation of the susceptibility map: FWO-Frontier Works Organisation, PMD-Pakistan Meteorological Department, GSP-Geological Survey of Pakistan, DEM-Digital Elevation Model.**

## 5.1  Literature Review

In the first step, existing information and data for the study area were collected from archives of the Frontier Works Organization (FWO), Geological Survey of Pakistan (GSP), Pakistan Meteorological department (PMD) and research catalogues (Khan et al., 2000). FWO is responsible for clearance and maintenance of the KKH after its potential blockage. Road maintenance log were collected and digitized. Following three important maps were prepared from data collected:

i.  A multi-temporal landslide inventory map (Fig. 6) was prepared using ArcGIS 10.3, based on GSP's publications (Fayaz et al., 1985; Khan et al., 1986, 2003) and road maintenance logs of FWO.

ii.    A comprising geological and seismo-tectonic map (1:50000 and 1:250000) of the area was prepared by digitizing and compiling various pre-existing maps (Khan et al. 2000). Data related to lithology, faults and seismicity has been extracted from these maps.

iii.    An annual precipitation map of the study area was prepared by using rainfall data of six weather stations and previous map (Fig.1 & Fig. 3) along the KKH.

### 5.1.1    Landslide Inventory

A landslide inventory map is an important instrument to display the location, date of occurrence and type of mass movement (Guzzetti et al., 2012).  Landslide inventory maps are prepared to define and record extent of mass movements in different regions, to investigate an impact of lithology, geological structures (fault, fold etc.) on types, distribution and occurrence of landslides, to use for preparation of landslide susceptibility mapping and to analyze geomorphic evolution of an area. Preparation of these maps involves multiple techniques based on satellite imagery, field interpretations and compilation of previous publications (Guzzetti et al., 2000; van Westen et al., 2006).

 In this study, we used real time data of landslide occurrences acquired from road clearance logs of Frontier Works Organization (FWO) for different periods (1982, 1983, 1996, 1997, 1998, 1999, 2000, 2014, 2015, 2016), publications of Geological Survey of Pakistan (GSP) (Fayaz et al., 1985; Khan et al., 1986, 2003), a research article (Hewitt, 1998), Google Earth imagery and field surveys to prepare multi-temporal landslide inventory along the Highway (Fig. 6). Polygon outlines for clearly visible landslides on satellite imagery (based on data of FWO and GSP) were drawn.  This landslide inventory map was then validated during field campaign. Spatial analysis and validation of the final susceptibility map were performed by using these polygons. Due to small scale of the inventory map, visibility of polygons was extremely low. Therefore, these polygons were then converted and displayed as points in the inventory map (Fig. 7).

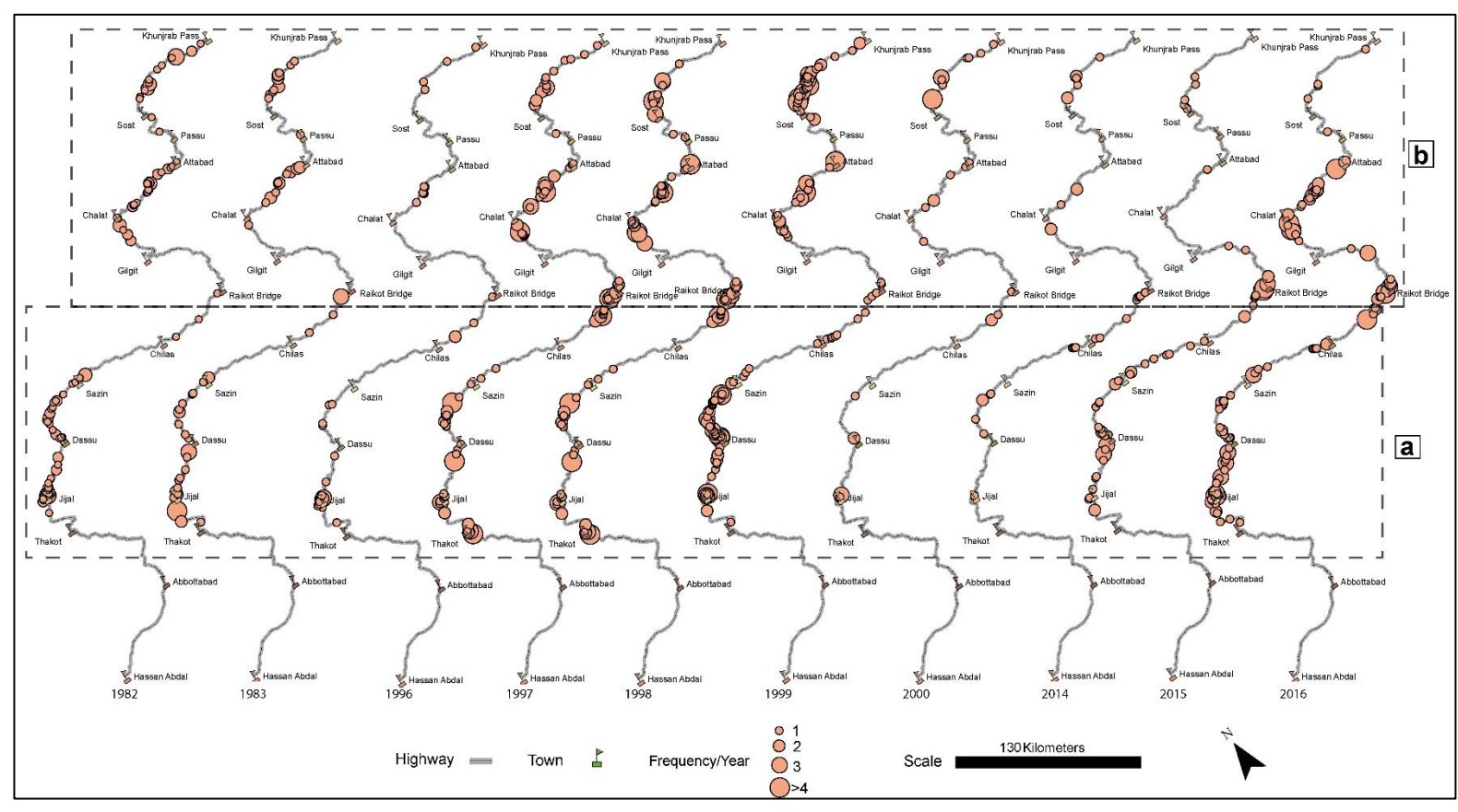

**Figure 6: Temporal distribution of landslides along the Highway: a and b represent two problematic section shown in Fig. 11.**

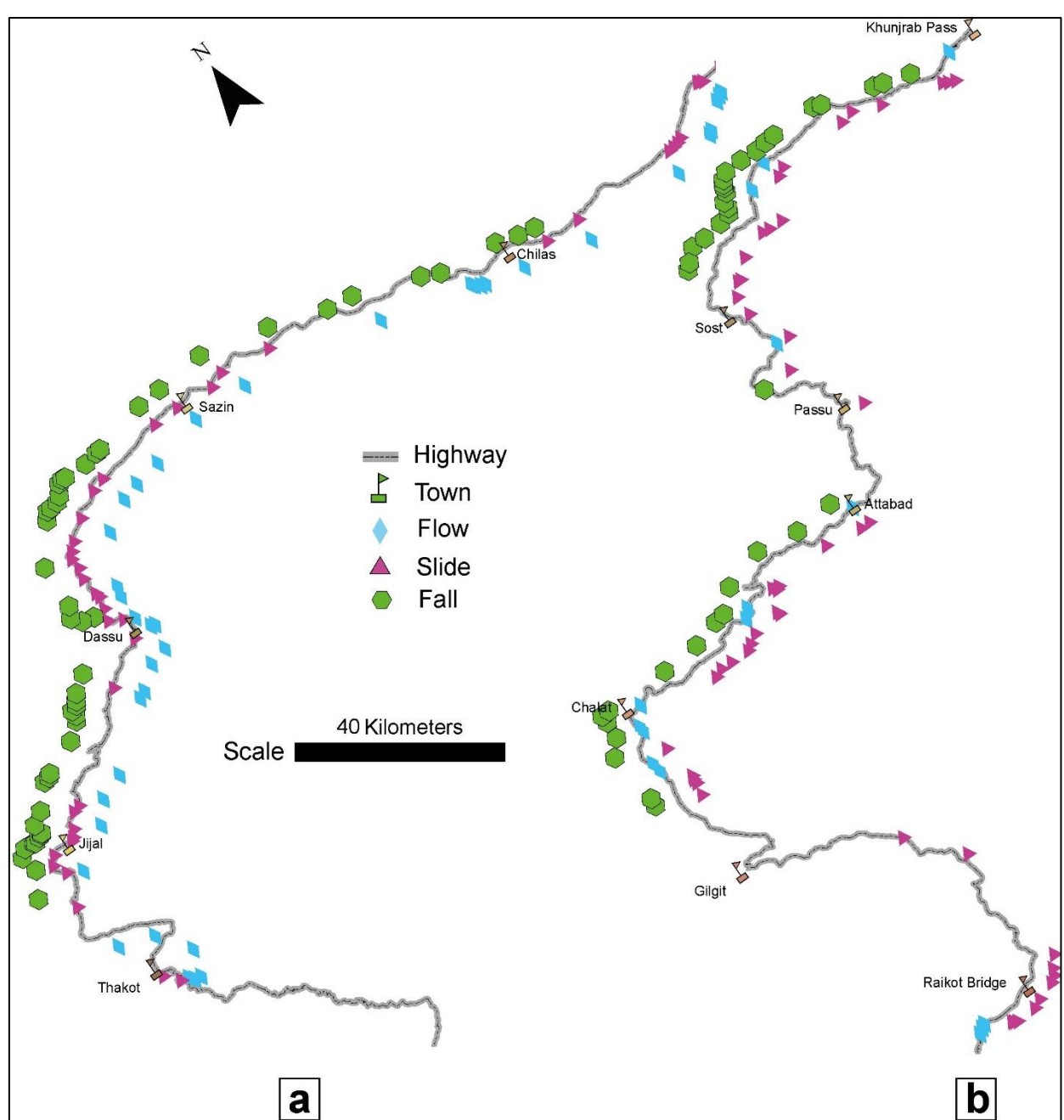

**Figure 7: Types of Mass Movements along the Highway (a) Thakot to Raikot Bridge (b) Raikot Bridge to Khunjrab Pass (Locations shown in Fig. 6)**

### 5.2    Field Observation

Locations of landslides, lithological contacts and faults were validated and supplemented during a field visit. In addition to locations, types, size, failure mechanisms and structural control of landslides were determined. Acquired data was further used to prepare landslide inventory map within 2 km$^2$ around the Highway.

### 5.3    Remote Sensing

Geomorphological parameters (elevation, slope, aspect, curvature) and drainage were extracted from Shuttle Radar Topography Mission (SRTM) based DEM (30 m $\times$ 30 m). Thematic layers were prepared and classified using the natural break method in ArcGIS 10.3. Satellite images of Landsat 8 (19-21 November 2017) were acquired from USGS web portal and then pre-processed by using QGIS 2.18's Semi-Automatic Classification Plugin (SCP), followed by supervised classification of composite images in Arc GIS 10.3to prepare land cover map.

#### 5.3.1    Construction of thematic maps

Thematic layers of elevation, slope, aspect and curvature were prepared and classified using the natural break method in ArcGIS 10.3. Drainage was extracted from SRTM based DEM by arc hydro tool. A buffer polygon of 300 meters was created to measure distance around streams to form thematic layer of distance from drainage. Faults, lithology and seismic zones were digitized from previously published geological and seismic maps. Multiple ring buffer polygons of 500, 1000, 1500 and 2000 meters around digitized faults were produced. Vector layers of distance from fault and drainage, lithology and seismic intensities were then rasterized. Annual rainfall data was interpolated to create precipitation map and then was combined with previously published annual rainfall map of PMD. Thematic layer of land cover was produced from land cover map.

### 5.4    Analytical hierarchy process (AHP)

Analytical hierarchy process (AHP) is a multi-criteria decision making approach to prepare landslide susceptibility maps. It has been used by previous researchers to assign weightage to landslide-controlling factors (Basharat et al., 2016; Kanwal et al., 2016; Shahabi and Hashim, 2015; Yalcin, 2008).   It is based on the user's decision to weigh factors through their pairwise comparisons. Each factor is assigned a score (1-9) depending upon its relative importance, with increasing impact from 1 to 9 (Table 2, Saaty, 1990). The values assigned are based on spatial analysis of data, field observations and experience of the user. If the parameter on the x-axis is more important than the one on y-axis, the value ranges between 1 and 9. Conversely, when the factor on y-axis is more important, the values are in reciprocals (1/2-1/9).

**Table 2: Fundamental scale for pair-wise comparisons (Saaty, 1987)**

| Intensity of Importance | Explanation |
|:---:|:---|
| 1 | Equal Importance |
| 2 | Weak or slight |
| 3 | Moderate importance |
| 4 | Moderate plus |
| 5 | Strong importance |
| 6 | Strong plus |
| 7 | Very strong |
| 8 | Very, very strong |
| 9 | Extreme importance |

Consistency ratio (CR) is a tool to check and avoid inconsistencies and bias in whole process of rating controlling parameters (Basharat et al., 2016; Kanwal et al., 2016; Komac, 2006; Pourghasemi and Rossi, 2017; Sarkar and Kanungo, 2004; Taherynia et al., 2014; Yalcin, 2008).

$$CR = CI/RI \qquad (1)$$

Where CR is consistency ratio, CI is consistency index and RI is random index.

CI was calculated by using following equation:

$$CI = (\lambda_{max} - n)/n - 1 \qquad (2)$$

Where $\lambda_{max}$ is the maximum eigenvalue of matrix and n is the number of controlling parameters involved (Zhou et al., 2016).

Saaty (1987) produced a table (Table 3) of random consistency index (RI) after calculation from 500 samples. Values of RI from this table and calculated CI were then compared to find CR.

**Table 3: Random Consistency Index (RI)**

| n | 2 | 3 | 4 | 5 | 6 | 7 | 8 | 9 | 10 |
|:---:|:---:|:---:|:---:|:---:|:---:|:---:|:---:|:---:|:---:|
| RI | 0 | 0.58 | 0.9 | 1.12 | 1.24 | 1.32 | 1.41 | 1.45 | 1.51 |

**Table 4: Pairwise matrix and weights of factor sub-classes.**

| Class | (1) | (2) | (3) | (4) | (5) | (6) | (7) | (8) | % | Importance |
|---|---|---|---|---|---|---|---|---|---|---|
| Aspect | | | | | | | | | | |
| (1) North | 1.00 | | | | | | | | 2.59 | 2 |
| (2) Northeast | 2.00 | 1.00 | | | | | | | 5.89 | 3 |
| (3) East | 3.00 | 2.00 | 1.00 | | | | | | 7.70 | 4 |
| (4) Southeast | 6.00 | 3.00 | 2.00 | 1.00 | | | | | 12.68 | 5 |
| (5) South | 7.00 | 4.00 | 2.00 | 1.00 | 1.00 | | | | 16.71 | 6 |
| (6) Southwest | 8.00 | 4.00 | 3.00 | 2.00 | 2.00 | 1.00 | | | 21.95 | 7 |
| (7) West | 9.00 | 5.00 | 4.00 | 2.00 | 3.00 | 2.00 | 1.00 | | 24.54 | 8 |
| (8) Northwest | 3.00 | 2.00 | 1.00 | 0.50 | 0.25 | 0.50 | 0.33 | 1.00 | 7.94 | 4 |
| Elevation (m) | | | | | | | | | | |
| (1) 432-1000 | 1.00 | | | | | | | | 4.05 | 1 |
| (2) 1000-2000 | 7.00 | 1.00 | | | | | | | 41.46 | 8 |
| (3) 2000-3000 | 6.00 | 0.50 | 1.00 | | | | | | 22.26 | 6 |
| (4) 3000-4000 | 6.00 | 0.33 | 1.00 | 1.00 | | | | | 20.73 | 6 |
| (5) 4000-4700 | 4.00 | 0.20 | 0.50 | 0.50 | 1.00 | | | | 11.50 | 4 |
| Slope | | | | | | | | | | |
| (1) 0-15 | 1.00 | | | | | | | | 3.92 | 1 |
| (2) 15-30 | 3.00 | 1.00 | | | | | | | 11.00 | 4 |
| (3) 30-45 | 9.00 | 4.00 | 1.00 | | | | | | 41.46 | 9 |
| (4) 45-65 | 8.00 | 3.00 | 0.50 | 1.00 | | | | | 30.83 | 6 |
| (5) >65 | 5.00 | 1.00 | 0.25 | 0.30 | 1.00 | | | | 12.80 | 4 |
| Land Cover | | | | | | | | | | |
| (1) Vegetation | 1.00 | | | | | | | | 9.68 | 2 |
| (2) Water | 0.33 | 1.00 | | | | | | | 4.75 | 1 |
| (3) Snow | 3.00 | 5.00 | 1.00 | | | | | | 23.08 | 4 |
| (4) Bare Rock/Soil | 6.00 | 9.00 | 4.00 | 1.00 | | | | | 62.49 | 7 |
| Rainfall Intensity (mm/y) | | | | | | | | | | |
| (1) 0-250 | 1.00 | | | | | | | | 5.68 | 1 |
| (2) 250-500 | 3.00 | 1.00 | | | | | | | 12.65 | 3 |
| (3) 500-1000 | 5.00 | 3.00 | 1.00 | | | | | | 27.35 | 5 |
| (4) 1000-1500 | 8.00 | 5.00 | 3.00 | 1.00 | | | | | 60.00 | 9 |
| Lithology | | | | | | | | | | |
| (1) Group A (SI, AF, HF, KgB, MG, TaF) | 2.00 | | | | | | | | 4.52 | 1 |
| (2) Group B (BeG, SC, TeF) | 4.00 | 1.00 | | | | | | | 8.14 | 2 |
| (3) Group C (KoB, KaB, CC, GilF, GJ) | 6.00 | 2.00 | 1.00 | | | | | | 14.83 | 4 |
| (4) Group D (KA, GirF, KF, RPV, SC) | 8.00 | 3.00 | 2.00 | 1.00 | | | | | 26.02 | 6 |
| (5) Group E (Qu, JC, OM, MS, PS, NPG, BaG) | 4.00 | 5.00 | 4.00 | 2.00 | 1.00 | | | | 46.49 | 9 |
| Seismic Intensity | | | | | | | | | | |
| (1) I-III | 1.00 | | | | | | | | 3.74 | 2 |
| (2) IV-V | 3.00 | 1.00 | | | | | | | 7.63 | 3 |
| (3) V-VI | 5.00 | 3.00 | 1.00 | | | | | | 14.22 | 6 |
| (4) VII-VIII | 7.00 | 5.00 | 3.00 | 1.00 | | | | | 29.77 | 8 |
| (5) IX-X | 8.00 | 6.00 | 4.00 | 2.00 | 1.00 | | | | 44.64 | 9 |
| Distance from a Fault (m) | | | | | | | | | | |
| (1) 0 -500 | 1.00 | | | | | | | | 50.50 | 9 |
| (2) 501-1000 | 0.50 | 1.00 | | | | | | | 27.04 | 7 |
| (3) 1001-1500 | 0.25 | 0.50 | 1.00 | | | | | | 15.30 | 5 |
| (4) 1501-2000 | 0.20 | 0.25 | 0.33 | 1.00 | | | | | 7.15 | 4 |

**Table 5: Pairwise matrix and weights of all controlling parameters.**

| Controlling Factors | Aspect | Elevation | Fault | Lithology | Land Cover | Distance from drainage | rainfall intensity | slope | curvature | seismicity | Weight |
|---|---|---|---|---|---|---|---|---|---|---|---|
| Aspect | 1 | | | | | | | | | | 1.77 |
| Elevation | 3 | 1 | | | | | | | | | 3.05 |
| Distance from Fault | 9 | 7 | 1 | | | | | | | | 23.28 |
| Lithology | 6 | 5 | 1/3 | 1 | | | | | | | 10.28 |
| Land Cover | 4 | 3 | 1/4 | 1/3 | 1 | | | | | | 7.04 |
| Distance from drainage | 6 | 5 | 1/3 | 1 | 2 | 1 | | | | | 8.90 |
| Rainfall intensity | 5 | 4 | 1/4 | 1/2 | 2 | 1/2 | 1 | | | | 7.08 |
| Slope | 9 | 7 | 1 | 3 | 5 | 3 | 4 | 1 | | | 23.74 |
| Curvature | 1 | 1/3 | 1/9 | 1/6 | 1/5 | 1/5 | 1/5 | 1/7 | 1 | | 1.82 |
| Seismicity | 7 | 5 | 1/3 | 1 | 4 | 2 | 3 | 1/3 | 7 | 1 | 13.03 |

The value of CR indicates the inconsistency in the expert's decision during weighing the parameters. Values of CR lower than 0.1 prove the decision consistent, while values greater than 0.1 indicate inconsistency and suggest a revision of judgement. Subclasses in each factor were prioritized by using pairwise comparison procedure (Table 4). Curvature and distance from drainage comprised of two and one classes respectively. Therefore, influence of these subclasses was easily scaled without AHP procedure. In next step, each parameter was assigned weightage on the completion of procedure (Table 5). Value of CR in our study remained below 0.1, which proves comparisons and weighting criteria reliable, unbiased and consistent.

## 5.5     Weighted Overlay Method

Weighted overlay method (WOM) is a simple and direct tool of Arc GIS to produce a susceptibility map (Bachri and Shresta, 2010; Intarawichian and Dasananda, 2010). Many researchers used WOM to produce landslide susceptibility map (Bachri and Shresta, 2010; Basharat et al., 2016; Intarawichian and Dasananda, 2010; Roslee et al., 2017; Shit et al., 2016). We used an overlay of raster layers of all controlling factors to prepare a susceptibility map. Raster layers of each controlling factor were reclassified and weighted according to their importance determined by AHP (Table 4 & 5). The cumulative weight of all input layers was maintained at 100. All layers were combined by using weighted overlay tool based on following equation (3):

$$S = \frac{\sum W_i \, s_{ij}}{\sum W_i} \qquad (3)$$

Where $W_i$ is weight of $i$th factor, $S_{ij}$ represents subclass weight of $j$th factor and S is spatial unit of the final map. The completion of this process resulted into the ultimate production of a landslide susceptibility map of the Highway.

## 6    Results

### 6.1    Landslides along the KKH

A total 261 landslides were used to prepare the map (Table 6). Broadly, we grouped these into shallow and deep-seated landslides (rock avalanches). Shallow landslides were further divided into slides, falls and flows based on simplified version of Varnes (1978).  Four sections (Table 6) are characterised by a large number of landslides during heavy rainfall and snowmelt. Highway blockage and traffic interruption in these sections is a regular phenomenon. Presence of a large number of rock/debris falls (37) in Jijal-Dassu section is due to steep topography formed by deep river incision in ultramfics (Jijal Complex), amphibolites (Kamila Amphibolites) and Gabbronorites (Chilas Complex) of Kohistan Island Arc. Whereas, stress release joints with short persistence in Sazin-Chilas section are responsible for huge boulder falls ($>6m^3$). A large number of slides (rock, debris and mud) and flows (debris and mud) in old large landslide deposits characterizes Raikot Bridge section. Hunza valley section is dominated by slides (rock, debris and mud) in highly sheared rockmass and falls (rock and debris) in over steepened parts of the valley. Steady flow of traffic along the highest section (Sost-Khunjrab Pass) of the Highway is a major problem due to seasonally influenced falls and slides. This section has a large number of large landslides (16), which dammed the Hunza and Khunjrab rivers in past.

**Table 6: Types of landslides along the KKH.**

| Section | Rock Falls | Flows (Debris, Mud) | Shallow Landslides | Deep-seated Landslides/Rock Avalanches |
|---|---|---|---|---|
| Jijal-Dassu Section (91 Km) | 37 | 17 | 19 | 1 |
| Sazin-Chilas Section (90 Km) | 10 | 9 | 7 | 4 |
| Raikot Bridge Section (49 Km) | 5 | 8 | 15 | 5 |
| Hunza Valley Section (76 Km) | 13 | 6 | 15 | 10 |
| Sost-Khunjrab Valley Section (86 Km) | 22 | 8 | 16 | 16 |
| Rest of the Highway (321 Km) | 2 | 2 | 14 | 0 |
| Total | 89 | 50 | 86 | 36 |

### 6.2    Causative Factors and Spatial Distribution analysis

Geological, morphological, seismo-tectonic, topographic and climatic factors are generally considered as landslide-controlling parameters (Kamp et al., 2008). The following ten causative factors were considered for preparation of the map: Lithology, distance from faults, seismicity, elevation, slope, aspect, curvature, land cover, rainfall intensity and distance from drainage. Size and distribution of the landslides varies locally, depending on the values of the parameters mentioned above. Thus, the creation of an accurate and precise GIS-based landslide susceptibility map is entirely dependent on the availability of data

related to controlling factors (Ayalew et al., 2004). Rockslides, debris slides, rock avalanches, rock fall, toppling, wedging, mudflows and debris flows are important landslide processes along the KKH.

### 6.2.1 Lithology

Time and type of slope failure is determined by the slope-building lithology. Each lithology is unique in terms of response to stresses and therefore features a particular susceptibility to potential slope failure (Vallejo and Ferrer, 2011). The KKH traverses a great variety of lithologies comprising sedimentary, igneous and metamorphic rocks. According to spatial analysis, Quaternary deposits, the Jijal complex and the Misgar slates exhibit the highest numbers of mass movement events (Fig. 8f).

### 6.2.2 Distance from faults

The Main Mantle Thrust (MMT), the Kamila Jal Shear zone (KJSZ), the Kamila Strike-slip Fault (KSF), the Raikot Fault, the Main Karakoram Thrust (MKT) and the Karakoram Fault are important structural features in close proximity of the Highway (Fig. 4). Landslides are concentrated along these active faults where rockmass is highly deformed (Ali et al., 2017). The fact that 54% of mapped landslides were found within a maximum distance of 1 km from these faults, while 69% were found within a 2km range, impressively substantiates the postulated strong control of structural features (Fig. 8e).

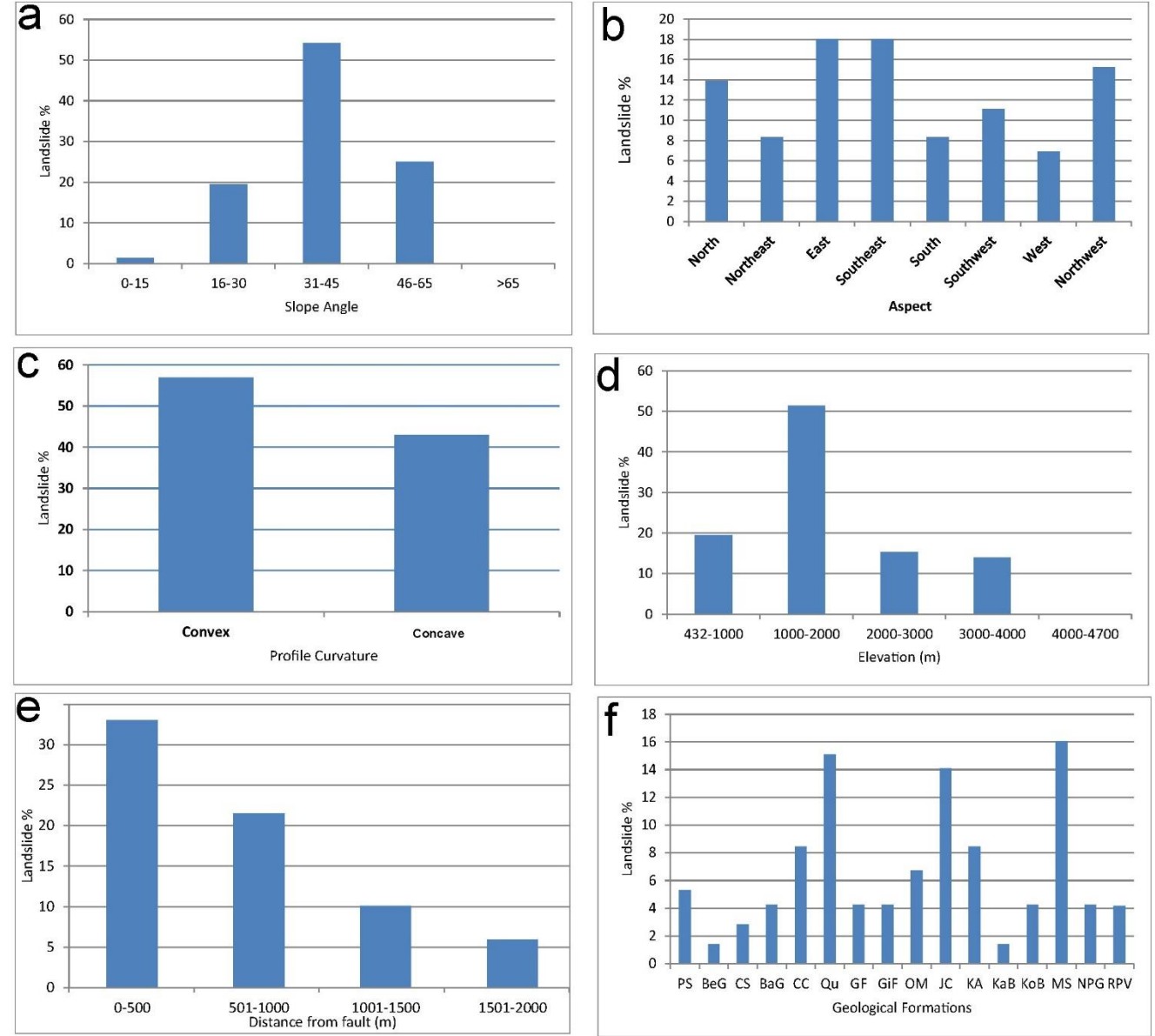

**Figure 8: Frequency distribution histograms of controlling parameters: a) Slope Angle b) Aspect c) Profile Curvature d) Elevation e) Distance from fault f) Geological formation (Abbreviations explained in Fig. 5). BaG: Baltit Group, BG: Besham Group, Cc:Chilas complex, GilF: Gilgit Formation, GirF: Gircha Formation, JC: Jijal Complex, KA: Kamila Amphibolite, KaB: Batholith, KoB: Kohistan Batholith, MS: Misgar Slates, NPG: Nanga Parbat Granitic Gneisses, OM: Ophiolitic Melange, PS: Passu Slates, Qu: Quaternary, RpV: Rakaposhi Volcanics, TF: Theilichi Formation, F: Fault, TF: Thrust Fault, MMT-Main Mantle Thrust, KJS-Kamila-Jal Shear Zone, KSF-Kamila Strikeslip Fault, RF-Raikot Fault, KF-Karakoram Fault, MKT-Main Karakoram Thrust.**

### 6.2.3 Geomorphologic Factors

Slope angle is an important geomorphic factor responsible for initiation of slope movements (Lee et al., 2004), to be considered for preparation of landslide susceptibility maps. Steep slopes are more susceptible to failure as compared to gentle ones. The study area demonstrates variation in topography ranging from steep to gentle slopes, high plains to narrow gorges and high cliffs. Slope steepness in the area has been divided into five classes. Division of slope steepness into classes was based on statistical analysis. Different classes were tried but found this division better in our study area. More than 50% of landslides occurred in class III (30°-45°) areas, whereas least landslide events (2%) occurred in class I and class V (0 - 15° and >65°) areas (Fig. 8a). In addition to slope and elevation, aspect and curvature were also considered important factors for preparation of the landslide susceptibility map. However, in our area these parameters seem to have a reduced influence on landslide occurrence (Fig. 8b, 8c, 8d).

### 6.2.4 Hydrology

The proximity of streams has been considered for the preparation of landslide susceptibility maps by many researchers (Akgun et al., 2008; Basharat et al., 2016; Kanwal et al., 2016; Wang et al., 2015). In our study area, many small tributaries feed main rivers, Indus and Hunza. During Monsoon season and after heavy precipitation events, these streams exhibit high-energy flows and large discharge, and are the main source of mud and debris flows. Heavy precipitation during monsoon and the westerlies triggers many landslides by increasing pore water pressure in unconsolidated sediments. Annual precipitation map of the area was prepared based on Pakistan Meteorological Department (PMD) data (Fig. 3a). A strong association between precipitation and mass movements along the Highway has been found (Fig. 3b ) (Ali et al., 2017). Peaks in mass movements curve is clearly synchronizing with high precipitation in respective months (Fig. 3b). A large number of landslides along the KKH occurred in 1999 leading to traffic blockade. Precipitation map was then overlaid to landslide events (1999). A large number of landslide events were found in south of Sazin with annual precipitation more than 1000 mm/y. Section of the KKH in East of Sazin contained comparatively less landslides due to its location in semi-arid to arid climate zones (>250 mm/y). Similar control of rainfall over landslide events has also been found in rest of the KKH.

### 6.2.5 Land Cover

Variations in land cover control the spatial distribution of landslides along with other conditioning parameters (lithology, seismology, slope geometry) (Malek et al., 2015). Changes in land cover influence the hydrological condition of the slopes, leading to slope instability. Generally, vegetation tends to resist the erosion process whereas bare rock or soil is more

susceptible to slope failure (Reichenbach et al., 2014). Restrepo and Alvarez (2006) found a strong relationship between land use and landslide events. Previous experts used a variety of softwares and techniques to produce a land cover map from satellite imagery. Many of them used Maximum Likelihood (ML) supervised classification tool on Arc GIS 10.3 (Ahmad and Quegan, 2012; Butt et al., 2015; Escape et al., 2014; Pourghasemi et al., 2012; Reis, 2008; Rwanga and Ndambuki, 2017; Ulbricht et al., 1993). All land cover maps produced by this technique had an accuracy more than 80%. Optical images of Landsat 8 (19-21 November 2017) were downloaded from USGS database. These images were then ortho-rectified and atmospherically corrected by using Semi-Automatic Classification Plugin (SCP) of QGIS 2.18. Composite images were classified by using Arc GIS 10.3's Maximum Likelihood (ML) supervised classification tool. Training data and spectral signature file was created to represent uniform four classes (Vegetation, Water bodies, Snow and Bare rock/soil). Produced map was divided into four classes: Vegetation, Water bodies, Snow and Bare rock/soil. The final land cover map was assessed using the confusion matrix method. Randomly distributed test pixels for each class were taken on the same image that had previously been classified. The accuracy of the land cover map added up to 87%. Due to variations in the mountain ecosystem from Hassan Abdal to Khunjab pass, the KKH is surrounded by vegetation, bare rock/soil, water bodies and ice/snow covered slopes. The section of the Highway between Hassan Abdal and Thakot is heavily vegetated due to considerable mean annual rainfall. From Thakot to Sazin, slopes are sparsely vegetated. From Sazin onward, barren rock slopes characterize the area.

In the end, spatial density analysis was performed to check influence of land cover changes on landslide events. Results revealed that more than 50% of landslide were located in bare rock/soil category whereas vegetation and snow-covered areas contain 23% each. Processed satellite images were captured at the start of winter season (19-21 November 2016). Most of the slopes in North of Gilgit were covered by snow at this specific time. It justifies presence of 23 % landslides in snow/glacial ice class.

## 6.3   Landslide Susceptibility Map

The produced landslide susceptibility map was classified in four classes: low susceptibility, moderate susceptibility, high susceptibility and very high susceptibility (Fig. 9 and 10). Nine (9) susceptibility levels were converted into four equally with interval of two except high susceptibility, which contains susceptibility levels of 5, 6 and 7. It was done to distinguish the locations that are more hazardous.

Areas of 49.9% and 10.4% of the classified map respectively belong to the high susceptibility and very high susceptibility classes, particularly owed to the presence of active faults, seismic zones and steep slopes (Table 7). 34.1% and 5.4% of the Highway fall into intermediate and low susceptibility areas, respectively.

**Table 2: Area of Susceptibility Classes**

| Classes | Area (%) |
|---|---|
| Low Risk | 5.4 |
| Intermediate Risk | 34.1 |
| High Risk | 49.9 |
| Very High Risk | 10.4 |

Owed to lucidity and space reasons, the final version of the map was divided into two parts: Sections (1) Hassan Abdal-Chilas section and (2) Chilas-Khunjrab Pass section (Fig. 9 and Fig. 10). The Highway section 1 until Thakot, is characterized by broad valleys and gentle slopes covered by vegetation and therefore falls into the low to intermediate susceptibility zones (Fig 9). Contrastingly, the following section north of Thakot, particularly close to Jijal, lies in high and very high susceptibility zones (Fig. 9a). Threads arise from the presence of the southern suture (MMT), poor rockmass quality, the active IKSZ and steep slopes. In the section from Pattan to Sazin, more than half of the Highway is at high susceptibility and very high susceptibility (Fig. 9b and 9c). This is because multiple shear zones (KJS) cross the highway between Pattan and Dassu and the surroundings of Sazin fall into the reach of Kamila strike slip fault (KSF) and the active HSZ (Fig. 9c). Two locations close to drainage features (Samar and Harbon Nala) near Sazin (Fig. 9c) also fall in the very high susceptibility zone.

The second section starts in Chilas and ends at the Khunjrab Pass (Chinese border). Some parts of the Highway were found at very high susceptibility (Fig. 10). The Raikot bridge section is the most dangerous part of the Highway as it lies directly over the active Raikot Fault (RF) and passes through RSSZ. Steep slopes and continuous erosion of slope toes by Indus River are aggravating the situation (Fig. 10a). Due to presence of the Northern suture (MKT), loose glacial deposits and steep slopes in the Hunza section, some locations of the Highway are declared very high susceptibility zones (Fig. 10b). Also north of Sost, two locations (Kafir Pahar and Notorious Killing zone) were found in very high susceptibility zones (Fig. 10c).

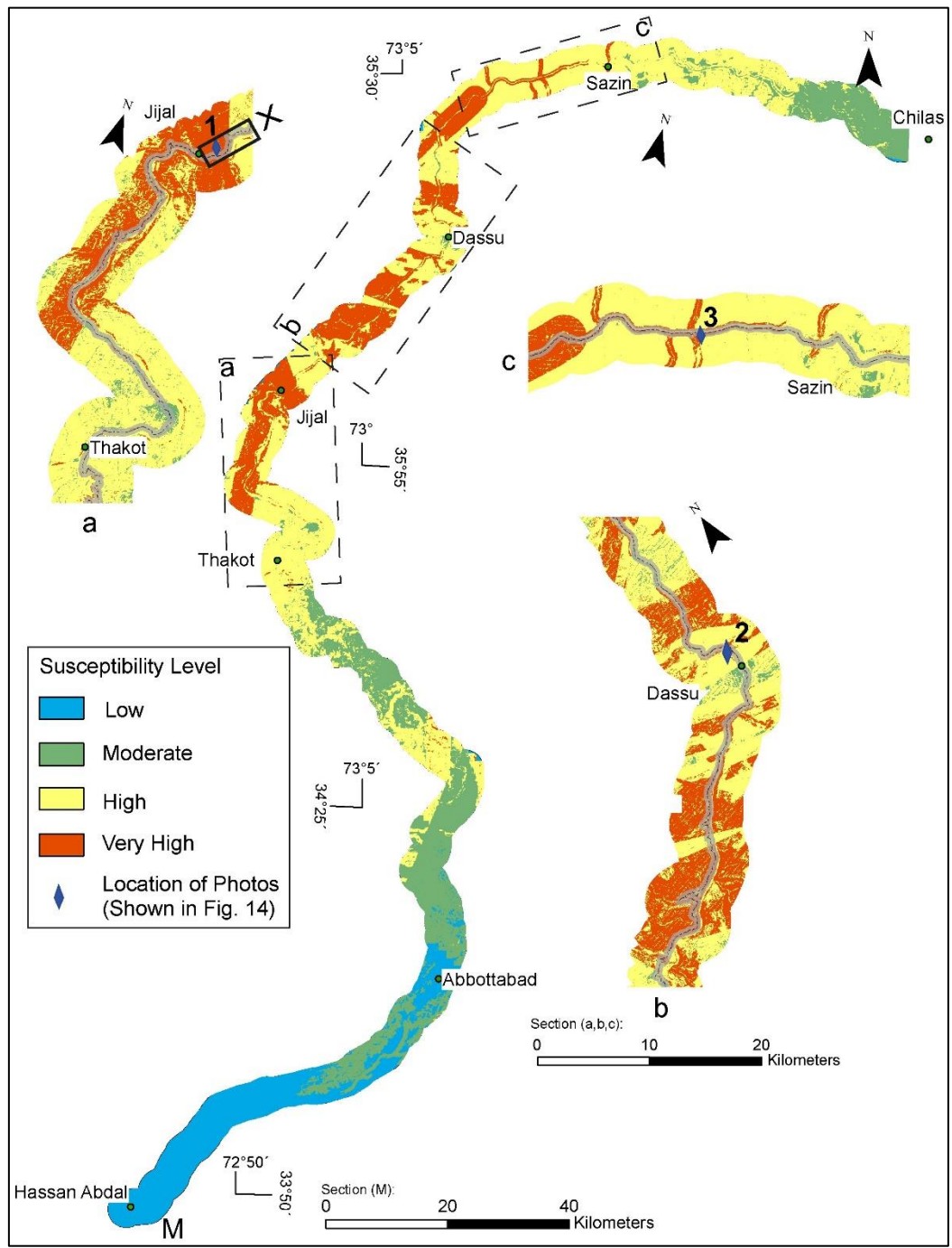

**Figure 9: Landslide Susceptibility Map (Abbottabad-Chilas): (a) Jijal Section (area in box "X" is shown in Fig. 15) (b) Dassu Section (c) Sazin Section.**

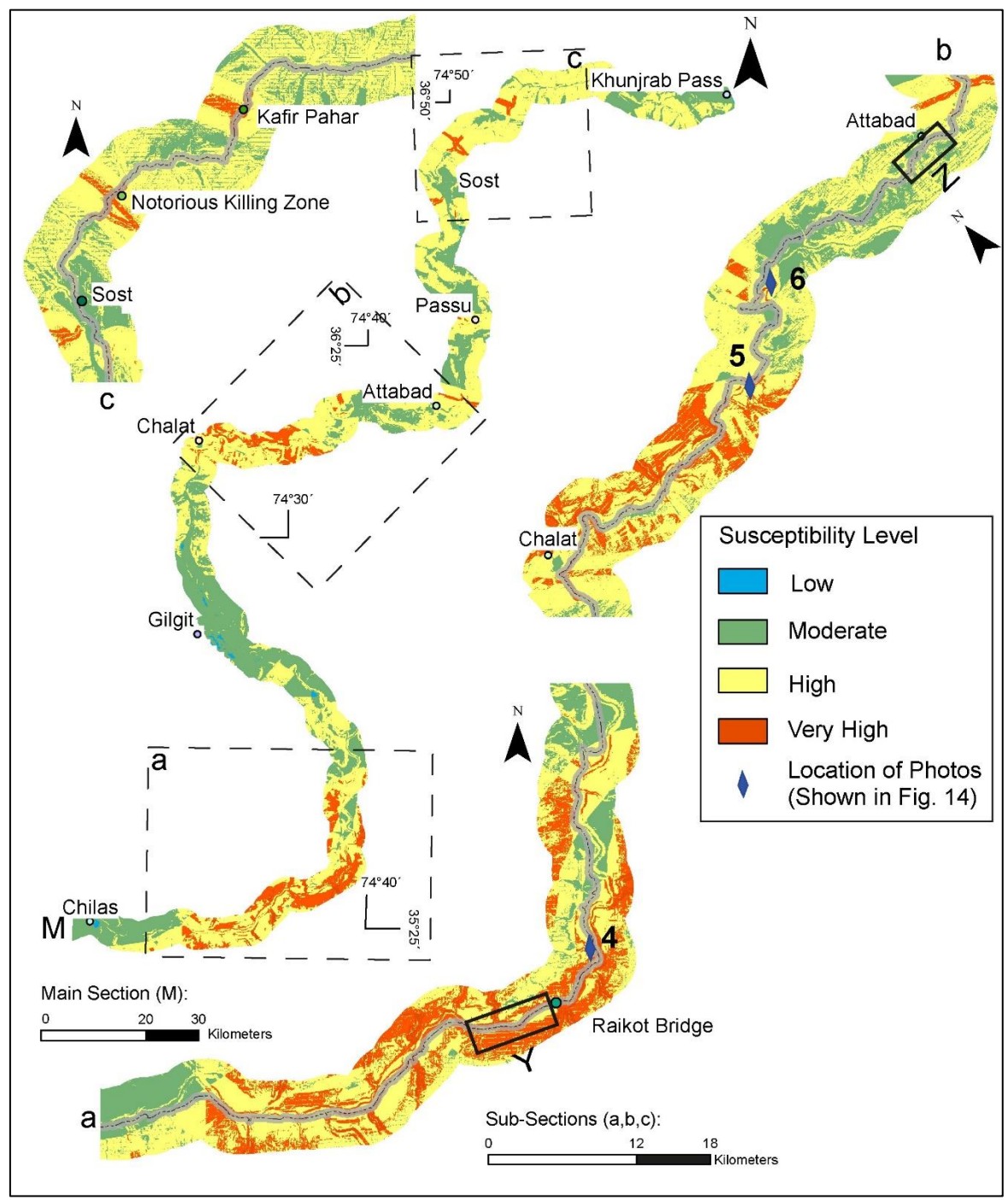

**Figure 10: Landslide Susceptibility Map (Chilas-Khunjrab Pass): (a) Raikot Bridge Section (box "Y" is shown in Fig. 16) (b) Attabad Section (box "Z" is shown in Fig. 17) (c) Sost Section.**

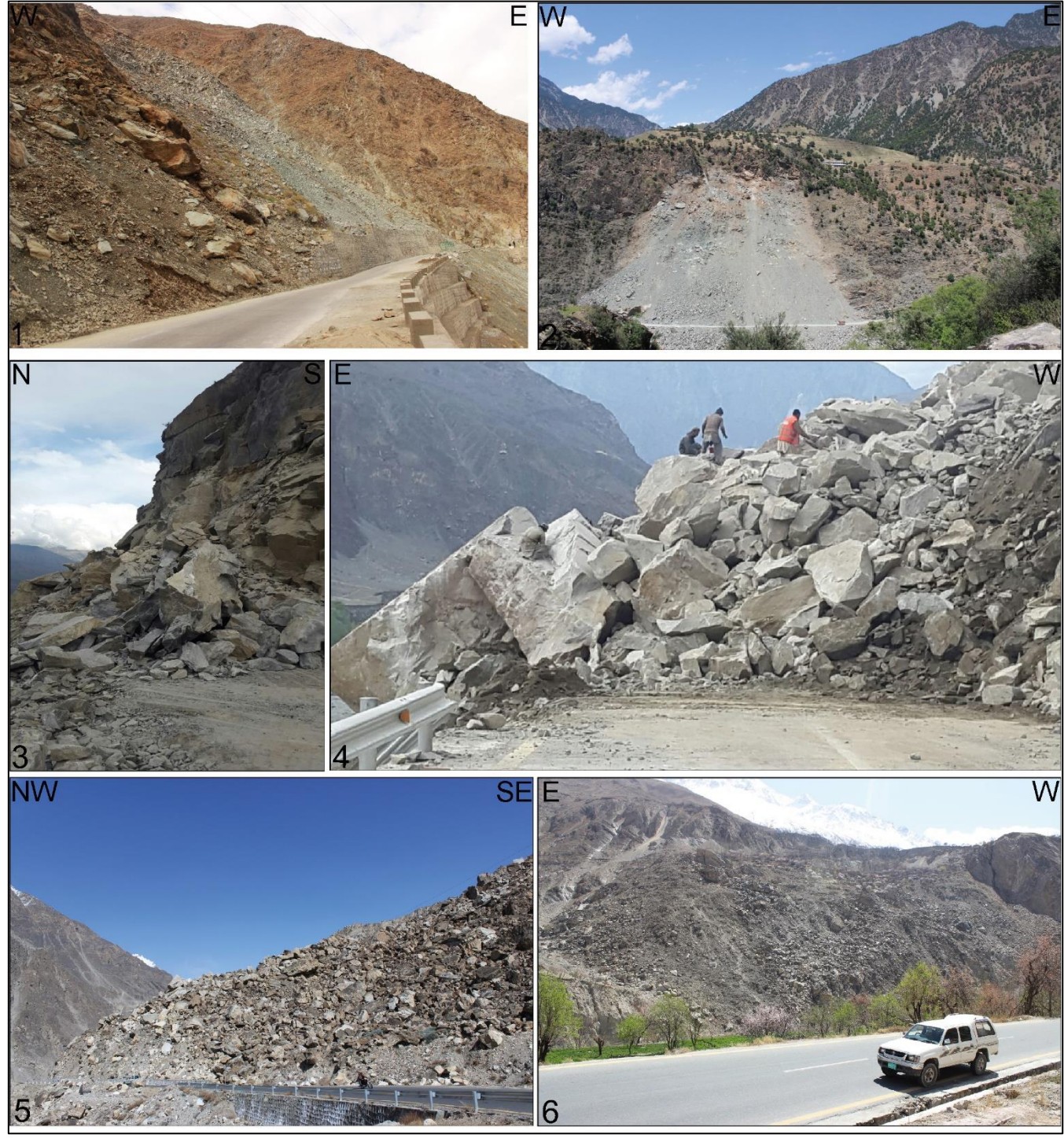

**Figure 11: Example of landslide events (Ali et al., 2017) (locations of the photos are given in Fig. 12 & 13)**

## 6.4    Accuracy assessment

Accuracy assessment of the map is an essential component of the whole process. In the past, different statistical techniques have been employed to check the predictive ability of a landslide susceptibility map: Predication Rate Curve (PRC), Landslide Density Analysis (LDA), Receiver Operating Curve (ROC) and Area Under Curve (AUC) (Deng et al., 2017). ROC is a better choice than other techniques as it is threshold-independent and measures both accuracy and error rate (Fawcett, 2006; Vakhshoori and Zare, 2018). Multiple researchers used ROC for validation of produced maps (Ahmed, 2015; Basharat et al., 2016; Lee, 2005; Zhou et al., 2016). In this study, we also used ROC and LDA to estimate the predictive accuracy of the map. In the first step, map classes were compared with landslide densities in their respective classes. Spatial analysis of the map and landslide events was performed on ArcGIS 10.3 using the tabulate area tool. According to the obtained results, most of the landslide events were found in high and very high susceptibility areas and very few landslides were present in moderate and low susceptibility zones respectively (Table 8). These statistics confirm a strong connection between susceptibility zonation and landslide events. Thus, this assessment indicates an adequate accuracy of the map.

**Table 3: Areas of susceptibility level of map and observed Landslides**

| Susceptibility Level | Area (km$^2$) | Landslides |
|:---:|:---:|:---:|
| 1 | 1.3 | 0 |
| 2 | 342.1 | 0 |
| 3 | 627.2 | 0 |
| 4 | 1517.1 | 1 |
| 5 | 1819.0 | 8 |
| 6 | 1316.1 | 29 |
| 7 | 585.3 | 20 |
| 8 | 67.4 | 13 |
| 9 | 0.2 | 1 |

In the second step, we used ROC for validation and accuracy assessment of the map, following the example of previous studies (Ahmed, 2015; Basharat et al., 2016; Brenning, 2005; Deng et al., 2017; Pradhan et al., 2010; Shahabi and Hashim, 2015). ROC is the product of a graphical plot opposing true positive rate (TPR) and false positive rate (FPR) (Fig. 12). TPR indicates the correctly predicted events and is plotted on the y-axis while FPR indicates falsely predicated events, and is plotted against the x-axis.  Area under curve (AUC) in a graphical plot explains the efficiency of the model. AUC may range from 0.5 to 1 in different cases depending on the accuracy of model. A value close to 0.5 indicates random results while values close to 1

indicate a perfect model (Ahmed, 2015). In this study, we used 72 landslide locations to validate the final version of the map. AUC was found 0.72 indicating a reputable accuracy (72%) of the map (Basharat et al., 2016; Deng et al., 2017).

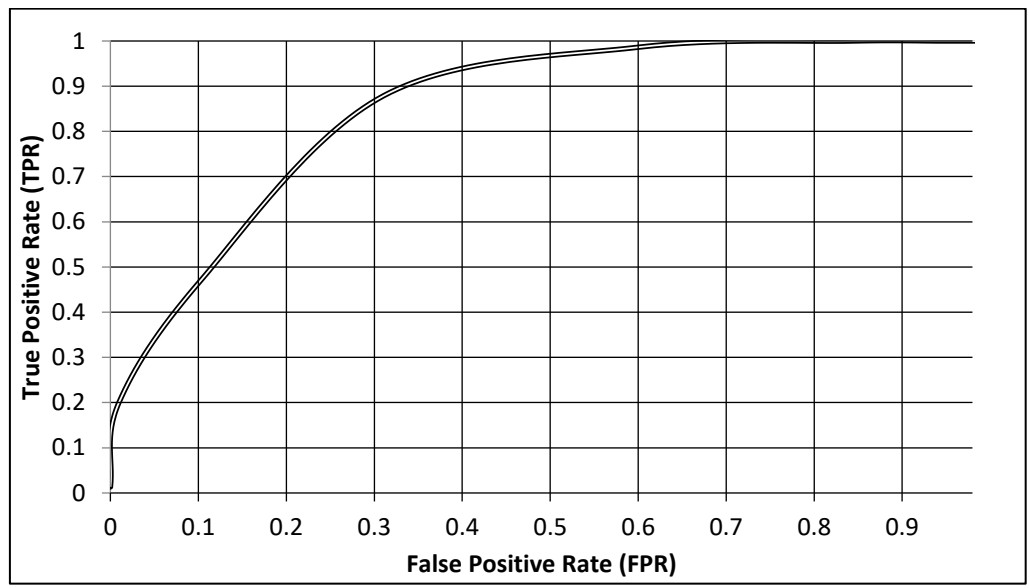

**Figure 12: ROC based accuracy assessment of the landslide susceptibility map**

## 7    Discussion

In this study, AHP based weighted overlay technique has been used to produce landslide susceptibility map along the KKH. Ten landslide controlling factors were considered for production of landslide susceptibility map. Review of previous published articles helped us to finalize causative factors, responsible for occurrence of landslides. Afterwards, spatial analysis was performed to prioritize and rate these parameters. The clustering of landslides in active fault and shear zones indicate their strong control. These tectonic features resulted into highly fractured and jointed rockmass, highly susceptible for failure. Slope gradient and distance from a fault were considered basic condition for slope instability and rated as the most important factor. Major seismic events in active seismic zones in close vicinity of the Highway, triggered landslides in the past. Some lithologies (slates, shales, quaternary) exhibit low shear strength and makes slope susceptible to failure. Therefore, seismicity and lithology are rated high but less important than slope gradient. Torrential Rainfall in Monson and Westerlies season triggers landslides and was taken into account. However, its influence decreases in north of Gilgit, where cumulative effect of ice melt with rise in temperature and rainfall cause landslides. Other geomorphic factors (elevation, aspect, curvature) were rated low because of their poor association with landslide events.

Quality of the data sets directly influences the quality of results (Fressard et al., 2014). Landslide inventory is the most important and basic one among all them. Inaccuracies regarding location and recent activity of landslides adversely affects accuracy of final map. We prepared and used landslide inventory by using satellite imagery, landslide activity log encompassing over ten years (Fig. 7) and then field surveys to validate locations and extent of landslides. The combination of

all these aspect has led into flawless inventory. We used geological maps (1:50000 and 1:250000) (Khan et al., 2000) explaining lithological variations within formation and also having both regional and local faults. Lithological contacts and location of faults and shear zones were verified during field visit. Seismic intensities and PGA (Peak Ground Acceleration) values were derived shake map of instrumental earthquakes (USGS). Rainfall data of six uniformly distributed weather stations

was used to prepare annual rainfall map. Land cover map was prepared from Landsat 8 optical imagery having 30-meter resolution. Images captured in November 2017 (before winter) were used to have snow free slopes along the KKH. Keeping in view the objective to produce landslide susceptibility map along the Highway, four classes (Vegetation, Water bodies, Snow and Bare rock/soil) were derived and temporal variability in landuse was not considered. In this study, quality of the data sets was comparatively better than previously used.

Many authors used AHP based model to prepare susceptibility map (Ahmed, 2015; Arizapa et al., 2015; Bachri and Shresta, 2010; Basharat et al., 2016; Intarawichian and Dasananda, 2010; Kamp et al., 2008; Komac, 2006; Park et al., 2013; Pourghasemi et al., 2016, 2012; Pourghasemi and Rossi, 2017; Rahim et al., 2018; Rozos et al., 2011; Shahabi and Hashim, 2015; Yalcin, 2008). Comparison of AHP based model with other models in some studies (Pourghasemi et al., 2012, 2016; Pourghasemi and Rossi, 2017; Shahabi and Hashim, 2015; Yalcin, 2008) proved former more accurate and precise. Accuracy

of the produced map in this study is 72% which is satisfactory and but slightly less than previous studies (Basharat et al., 2016; Kanwal et al., 2016; Pourghasemi et al., 2012; Pourghasemi and Rossi, 2017; Shahabi and Hashim, 2015; Yalcin, 2008). AHP is a simple and easy way to rate different parameters consistently. Value of CR remained below 0.10 for each case, indicating appropriate and reliable weighting criteria. However, AHP based model has been criticized due to its expert opinion based subjective approach. Therefore, we did spatial analysis to rate all controlling parameters to minimize chances of errors related

to cognitive limitations of expert. Further, due to low spatial resolution of DEM (30m x 30m), some of cut slopes along the KKH were neglected. Rating system introduced in this study may not fit into any other regions due to variations in geological, seismic, hydrological and other controlling parameters. Lastly, change in existing condition (undercutting of landslides toes for Highway expansion) of landslide controlling parameters may change present susceptibility along the Highway.

## 7.1    Case study

To supplement results and final landslide susceptibility map, three sub-sections were discussed: Jijal sub-section, Raikot Bridge sub-section and Attabad sub-section.

### 7.1.1    Jijal sub-section

Part of the Highway N of Jijal town lies in a zone of very high susceptibility (Box of Fig. 9a). It comprises highly fragmented ultramafic rocks of the Jijal complex. Due to its position in the hanging wall of MMT, it is highly jointed and locally sheared.

The area is seismically active and located just three kilometers away from the epicentre of the Pattan Earthquake on 28 December, 1974 (M = 6.2, D = 22 km). The seismic intensity (Modified Mercalli Intensity Scale) of this event along this part of the Highway reached VIII (Ambraseys et al 1981). Furthermore, the S of this area is seismically very active (Fig. 4c). During

the catastrophic October 2005 Kashmir Earthquake (M=7.6), some landslides were reactivated, leading to a closure of the Highway. Topography in this part is steep, with slope angles ranging between 40° and 70°. The area lies in the Monsoon region where average annual rainfall exceeds 1000 mm. The dotted yellow lines (Fig. 13a) indicate a big catchment area (1.34 km$^2$), capable of collecting large amounts of water during rainfall, leading to debris flows and debris slides in sheared and highly fragmented rock mass. Rock and debris falls are further promoted by clayey soils that form in joint apertures as a result of serpentinization. Due to heavy rainfall (617 mm) in March and April 2016, a large number of landslides was reactivated leading to blockage of the Highway for two weeks.  All these factors (closeness to fault, high seismicity, fragmented rock mass, heavy rainfall, steep topography) are responsible for the very high landslide susceptibility in this area.

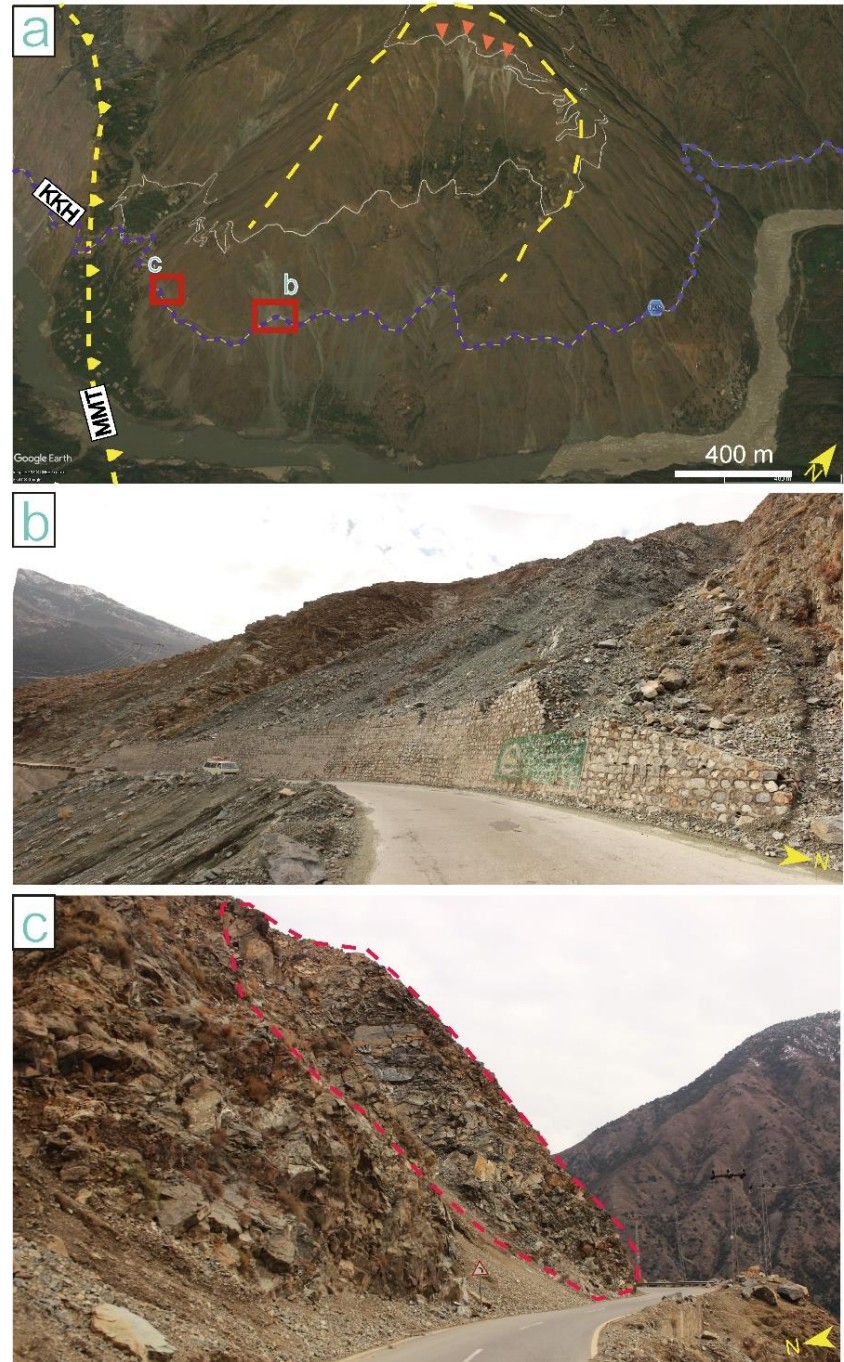

**Figure 13: Very high landslide susceptibility along the KKH near Jijal (Google Earth, 2017, Bing Maps) (for location see Box "X" in Fig. 12a). (a) Overview of 7.5 km long small section of the Highway in NE of Jijal: Dotted black line represents MMT; dotted yellow line marks the boundary of catchment area (1.34 km2); Pink arrows showing ongoing rock/debris falls, which travel downslope along with water during heavy rainfall; gully erosion is prominent. (b) Famous "Shaitan Pari Slide" with partially damaged retaining wall. Reactivation of this slide is mostly during heavy rainfall. (c) Highly jointed rockmass is highly susceptible to rock/debris fall.**

### 7.1.2   Raikot Bridge sub-section

Stability of the Highway is a challenge for geologists, civil engineers and Highway authorities. The Highway subsidence due river undercutting and presence of active Raikot Fault (RF) is a continuous phenomenon. Hot water springs and a shear zone (c. 125 m wide) indicate the presence of RF. It marks the boundary between Precambrian granitic gneisses of the Indian plate and batholiths and gabronorites of the Kohistan Island Arc. RF is responsible for shallow seismicity along the Highway (Fig. 4c). Seismic intensity (Modified Mercalli Intensity Scale) in this part reaches VI. The rockmass is highly jointed and sheared due to presence of the fault. Furthermore, the continuous seepage from hot water springs results in weathering and a lower shear strength of the rock mass. In the past, two large landslides dammed Indus River (Fig. 14a). Deposits of these landslides contain retrogressive slope movements and debris flows (Fig. 14a). Landslide damming had several effects on terraces and slopes, including the deposition of alluvial and lake deposits. In addition to this, continuous rockfall is adding large quantities of debris to the slopes. Topographically, also this section is very steep. Climatically, it lies in a semi-arid to arid zone with an average annual rainfall of 0 – 250 mm. Rainfall, however, is restricted to a couple of events per year. On 3 and 4 April, 2016, 105 mm rainfall reactivated debris flows and slides. The prevailing circumstances make this part of the Highway highly susceptible to landslides.

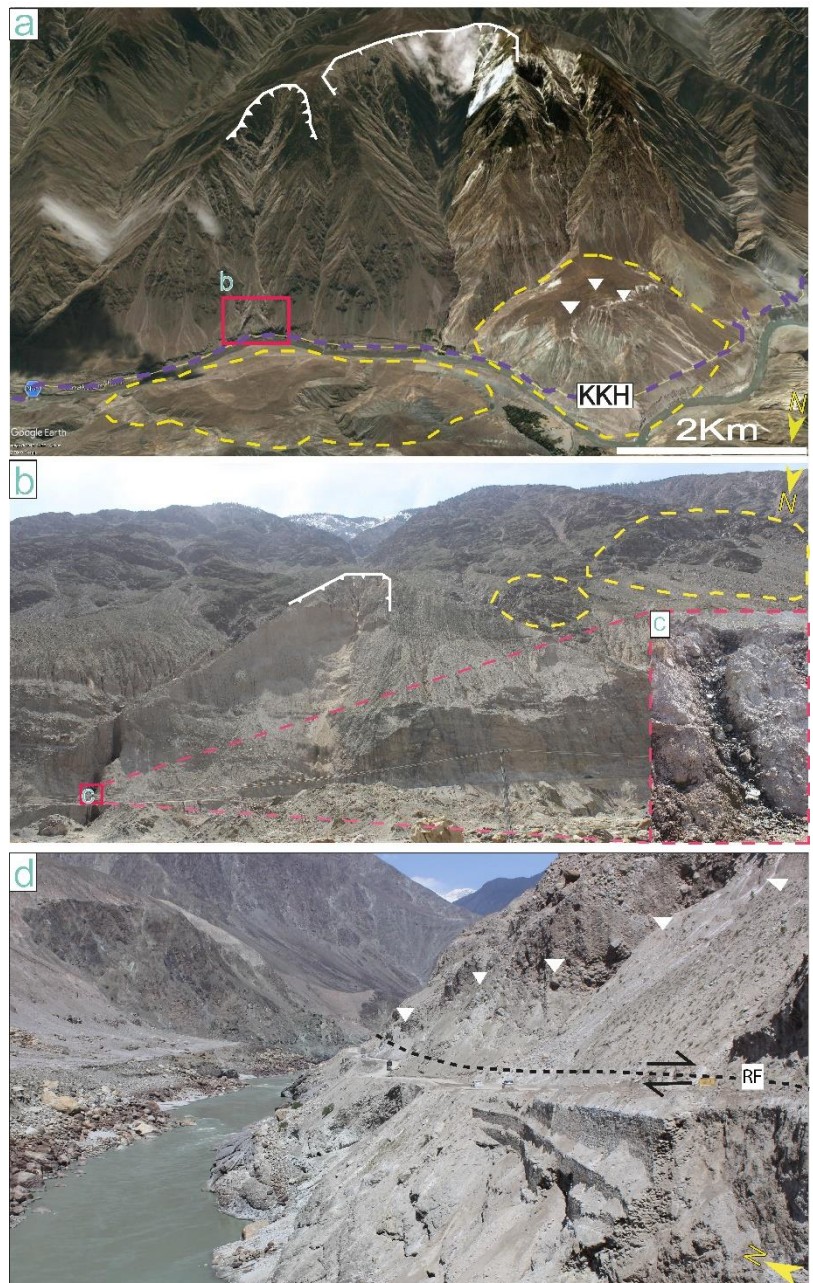

**Figure 14: : Very high landslide susceptibility near Raikot Bridge (Google Earth, 2017, Bing Maps) (for location see Box "Y" in Fig. 13b) (a) Overview of 13.4 Km long small section in West of the Raikot Bridge: White lines represent main scarps of large landslides/rock avalanches, which dammed the Indus River past; dotted yellow circles represent deposits of these old landslides; white arrows are showing sagging in landslide deposit which is due to retrogressive rotational failure (b) White lines represents scarp of shallow landslide in alluvial deposits (area in red box of a); area in dotted yellow circles represents ongoing rock/debris fall supplying scree/talus for debris flow during rainy season (c) One of the hot water springs (90oC-96oC) along the Highway in this section (d) Another view of small section: white arrows are marking upper limit of the shear zone (c. 125 m wide); dotted black representing active RF marked along shear zone; overhangs above and river erosion below the Highway makes this section highly susceptible to slope failures**

### 7.1.3    Attabad sub-section

Hunza section has a variety of slope failures depending upon prevailing geological and climatic conditions. However, the area shown in Fig. 15 is characterized by falls (rock, debris) and some slides (rock, debris). Historically, large landslides (1858, 1980, and 2010) dammed Hunza River in this part (Fig. 15a). The area is characterized by highly weathered and jointed granodiorites, orthogneisses and pegmatitic veins of the Kohistan Batholith. The orientation of joints (dipping in the same direction as the slope) has an adverse impact on slope stability. The area is located in the hanging wall of MKT, the main fault in this region, and a local fault exists in close vicinity (Fig. 17a). Past earthquakes (Astore, 2002; M=6.3, Muzaffarabad, 2005; M=7.6) produced ground shaking intensity up to V-VI (Modified Mercalli Intensity Scale). Climatically, the valley floor is part of a semi-arid zone (250 mm/yr – 500 mm/yr) while the higher slopes and peaks (> 5000 m) receive precipitation ≥ 1000 mm/yr. Therefore, the area is sparsely vegetated. Rainfall coupled with snowmelt in early spring reactivates old landslides (Ali et al., 2017). Undercutting of landslide toes by Hunza River below the Highway and rock/debris fall upslope are major concerns in this section. All of the above mentioned circumstances yield a high landslide susceptibility for this section (Fig. 11)

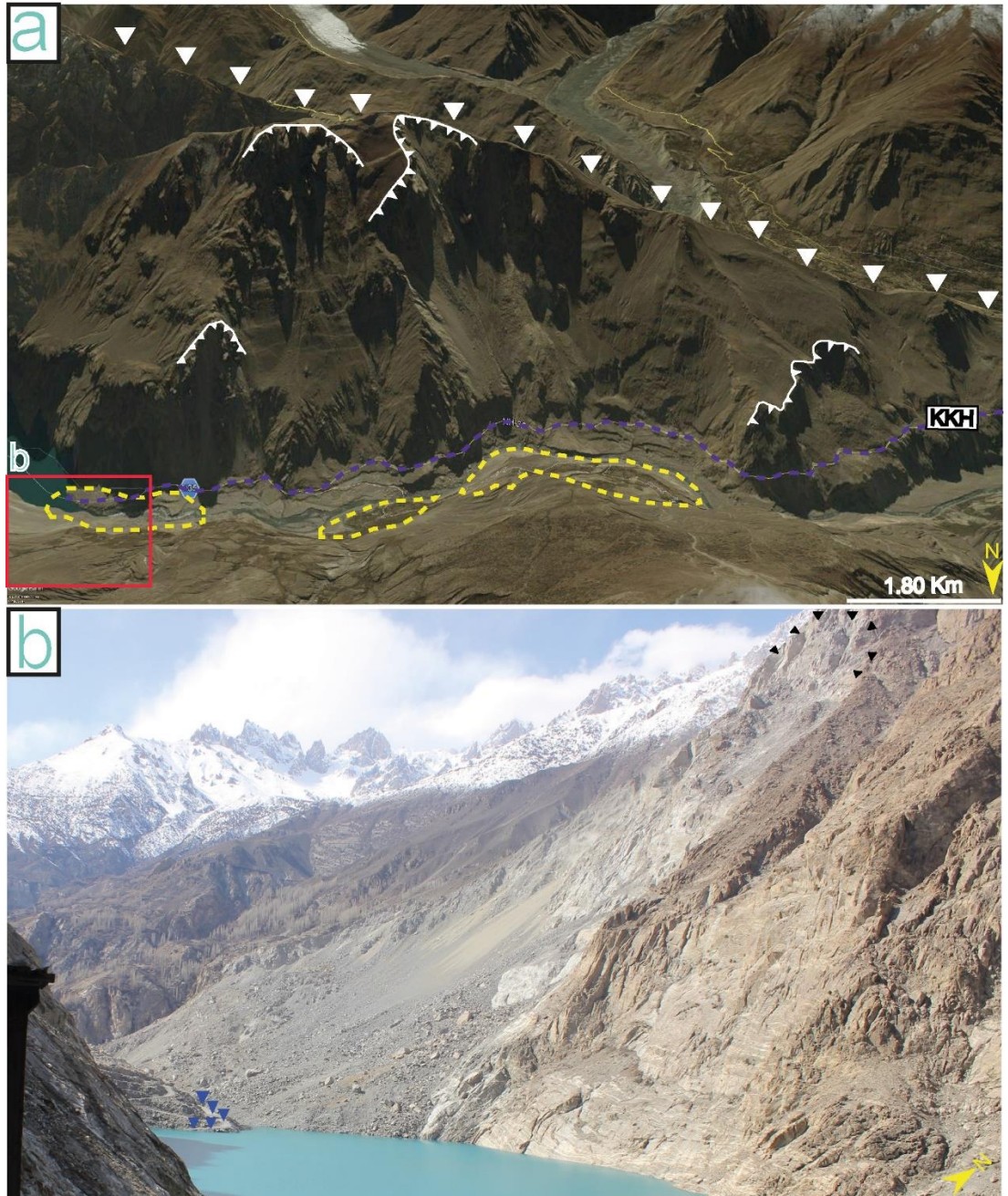

**Figure 15: High landslide susceptibility along the Highway in Hunza Valley (Google Earth, 2017, Bing Maps) (for location see Box "Z" in Fig. 13b) (a) Overview of 11 km long small section in west of the Attabad Lake: white lines are representing the main scarps of old large landslides/rock avalanches which dammed the Hunza River in past; dotted yellow circles represent deposits of these old landslides; white arrows are showing location of local fault (b) Attabad landslide (in red box of a); red arrows are showing the Highway submerged in lake water.**

# 8    Conclusions

A set of Landslide Susceptibility Maps of the KKH (CPEC) was prepared using ArcGIS 10.3 involving multiple techniques: literature review, remote sensing, field surveys. Ten controlling parameters (lithology, seismicity, rainfall intensity, distance from fault, elevation, slope angle, aspect, curvature, land cover and hydrology) were considered of which each one was assigned a numerical weight using AHP. Thematic layers of these parameters were then overplayed using the WOL tool of ArcGIS 10.3. Four different classes of landslide susceptibility were then applied to the final map: low susceptibility, moderate susceptibility, high susceptibility and very high Susceptibility. 10 % and 50% of the Highway were found in very high and high susceptibility zones, respectively.  Active faults (MMT, KJS, KSF, RF, MKT, KF), seismic zones (IKSZ, HSZ, RSSZ) and steep slopes are responsible for the associated susceptibility in these areas. The Highway is characterized by a variety of mass movements: rockfall, debris fall, rockslide, debris slide, debris flows and mudflows. The threatened sections are especially unstable in Monsoon and Westerlies seasons every year. Altogether, a detailed investigation is inevitable to enable hazard free and safe traveling. About 40% of the Highway lies in low and moderate susceptibility zones, which remain almost stable throughout the year. The Highway sections from Hassan Abdal to Thakot, near Chilas, Gilgit and Sost fall into these moderate and low susceptibility zones and are quite stable due to their course in broad U-shaped valleys with gentle slopes. Although the part of the Highway between Hassan Abdal and Thakot receives heavy precipitation in Monsoon season, the area is stable owed to mature geomorphology. Due to closeness with MMT, higher seismic intensity and steep topography, the KKH near Raikot Bridge and Jijal was found at very risk. Furthermore, extreme weather conditions, highly shattered and weathered rockmass, active faults and long steep slopes are responsible for very high and high susceptibility around Attabad, notorious Killing Zone and Kafir Pahar sites (Fig. 9 & 10). According to results, active faults, slope gradient, seismicity and lithology have a strong influence on landslide events along the Highway. In the final step, the predictive accuracy of the map was determined by using LDA and ROC. The accuracy of the map was rated to a satisfactory 72%, which is suitable for mitigation planning.

# 9    Acknowledgements

We would like to thank Higher Education Commission (HEC) of Pakistan and German Academic Exchange Service (DAAD) for financial support for the research project and Dr. Muhammad Basharat and Mr. Aram Fathian Baneh for their valuable suggestions. Further, we would like acknowledge Frontier Works Organization (FWO) and Pakistan Meteorological Department for provision of data for analysis

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
