# Peer review of "LANDSLIDE SUSCEPTIBILITY MAPPING BY USING GIS ALONG THE CHINA PAKISTAN ECONOMIC CORRIDOR (KARAKORAM HIGHWAY), PAKISTAN."

_Natural Hazards and Earth System Sciences, 2018_

## Referee Comment (RC1) · Anonymous Referee #1 · 20 Apr 2018

The manuscript "Landslide Susceptibility Mapping By Using GIS Along The China Pakistan Economic Corridor (Karakoram Highway), Pakistan", submitted by Sajid Ali and his co-workers for publication in the journal Natural Hazards Earth System Sciences, presents a GIS-based landslide susceptibility map of the Karakoram Highway, which connects China and Pakistan. This important highway is vulnerable to landslides, which may block the road and destroy the infrastructure. So the attempt to set up a landslide susceptibility map in this region is a valuable contribution to the hazard assessment around the Karakoram highway and may improve the maintenance and security along the road. But before publication in a scientific journal some major revisions seem to be needed, as originality and discussion of the data need to be specified and elaborated

in more detail.

First of all, the structure of this manuscript seems clear but some parts are missing or not well-balanced. Almost eight out of ten article pages, i.e. 80% of the written text deal with introduction, situation and methodology. The most important parts of a scientific article, results and discussion, fit in only one article page. In fact, the discussion chapter is missing at all, although some statements are "hidden" in the chapter "accuracy statement" and in the conclusions. This structure needs to be changed, and most important, a well-balanced discussion of the results has to elaborated. Please review your text passages thoroughly and sort primary information, results and discussion and clarify what the model outcomes mean scientifically.

In the following, I will comment on certain text passages and indicate questions. Introduction: I would appreciate more references on conditioning and triggering factors, and these can be used in the discussion again. Already in the first chapter, the insecurity of AHP is announced (p.2, line 29), and a combination with qualitative approaches and the use of GIS is considered a better option for regional studies.

General situation of the study area: How significant is the value of average annual precipitation in a Monsoon climate, also with abrupt changes to semi-arid / arid conditions? Are there more precise rainfall data available or can more datasets be taken into account in order to reduce the error source? Please clarify, how the precipitation data is used for the model.

Geology along the KKH: "Highly active landslide zones were identified from the distribution of existing landslides" – what about other zones which may have not been activated yet? Actually you do not know only based on the distribution of visible deposits how active a distinct zone is at the moment or at what certain frequency and what type of trigger is the main causing factor of the landslides. There might be lots of "old" deposits summing up to a higher total number of landslides in a rather "inactive" zone compared to a recently activated zone with less, but young landslides. Please
clarify how you mapped and characterized the landslides in the field. An attribute table of the landslides would be very helpful and support the discussion.

Seismology: The earthquake events in the region seem to be well-documented. How do you include these data in your hazard assessment? Especially the recent strong ones (e.g. the referenced in Oct 2005 or Oct 2015) could be used in a detailed case study. I would strongly recommend to do a detailed case study for at least one strong landslide event including lithology, seismology, conditioning and triggering factors. This would ideally fit your GIS-based mapping results. If not, this would also need to be discussed in detail.

Geomorphologic factors: How did you divide slope steepness into classes? I wonder on what criteria you based the data processing.

Literature review: Do you need this subsection? p.7, line 8: six weather stations along the highway: where are they (what climate?) and how does the rainfall influence your model? Please discuss, although (or because??) you classify the susceptibility levels based on active faults, seismic zones and steep slopes.

Field reconnaissance: These data sets should be presented in more detail. A landslide inventory map with classification and indication of magnitude and/or frequency and a case study could improve this paper a lot. So far only the location (Fig. 1) and the number (72) of the mapped landslides are given. For instance, you could set up tables like the colleagues T. Stanley and D. Kirschbaum (2017) in their study on global landslide susceptibility mapping (Table 4 and Table 5).

Remote sensing: DEM-quality of 30x30 m$^2$ and its influence on the results needs to be discussed, as several recent studies have shown that the DEM quality is crucial for modelling outcome. The accuracy of the land cover map of 87% needs to be discussed – is it a good or bad value compared to other studies?

Weighted overlay method: crucial part of this study. You should disclose the criteria.

In Table 2 the reader cannot follow your working procedure and how you chose the weighting factors.

Results: You need to explain the classification into four classes. What does a certain susceptibility level stand for? For this, table 3 could be developed in more detail (presence of active faults, seismic zones, steep slopes etc.).

Accuracy assessment: Please clarify, how the 72 landslide locations were used for validation. Only the location? What does the value of 72% tells us? Please discuss this outcome.

Chapter discussion is missing!

Conclusions: Primary information, for instance the classification of landslides into rock-fall, debris slide etc., must be provided earlier in the results (landslide map, attribute table as suggested above). Furthermore, some interpretation is mixed in here (stable/quite stable parts of the highway). This interpretation and discussion is very important for the paper. You should lay out a new chapter, as indicated above, including limitations of analysis and sources of error.

Table 4: Please indicate the absolute values of area and number of landslides because this improves the transparency of your data processing. Why/ based on what criteria did you reduce the number of 9 susceptibility levels to 4 in your final map? Please clarify.

Figure 1: Please indicate subfigures a), b) and c) and legend. Too many (bold) lines in mid-zoom map.

Figure 3: Scale in W-E direction is missing.

Figure 4b: Why don't you also focus on the earthquake zone in the south of highlighted section a)? Not only lithology but also seismic situation should be considered here!

Figure 5: Where is the highway? Please improve visibility.

Figure 6: a) How would your model change, if you split the group of 31-45° again? Why do you show the group >65° and in subfigure d) the group 4000-4700m? Is the y-axis "landslide %" based on all 72 landslides? Please clarify. Please use these values in the suggested landslide map/attribute table (case study) and compare with other studies.

Figure 10/11: Captions of subfigures a), b) and c) are missing. One of these recent events might be suitable for a detailed case study.

---

## Referee Comment (RC2) · Anonymous Referee #2 · 25 Apr 2018

After reviewing the manuscript entitled "*Landslide susceptibility mapping by using GIS along the China Pakistan Economic Corridor (Karakoram Highway) Pakistan*", following things were observed.

Authors performed landslide susceptibility analysis by applying analytical hierarchy process and weighted overlay method (WOL) in Karakoram Highway Pakistan. Authors used 10 landslide controlling factors to produce landslide susceptible map over the study area.

Generally, the issue considered in the manuscript is very important to Earth sciences community and could be publish in Earth System Sciences. However, the manuscript does not have the high scientific level presented by the journal.

The issue considered in the manuscript has been deeply discussed for decades, with many papers, methodologies and case studies. New works are more than welcome as long as they contribute something new to Earth science community, which may be new approaches, new case studies, and of course an improvement over the methods already published. Unfortunately this paper, even if engaging, doesn't offer anything new.

Here goes the list of critical shortcomings:

-Generally, WOL method presented in this study was firstly presented in 1994 in work :

Bonham-Carter, G.F., 1994. Geographic information systems for geoscientists: modelling with GIS. Pegamon Press, Oxford.

Since this time a lot of improvements, strategies, combination, comparison with other techniques have been presented in scientific community, the method is not new. Please check this papers with novelties in similar methodologies:

- Yalcin, A. (2008). GIS-based landslide susceptibility mapping using analytical hierarchy process and bivariate statistics in Ardesen (Turkey): comparisons of results and confirmations. *Catena*, *72*(1), 1-12
- Althuwaynee, O. F., Pradhan, B., Park, H. J., & Lee, J. H. (2014). A novel ensemble bivariate statistical evidential belief function with knowledge-based analytical hierarchy process and multivariate statistical logistic regression for landslide susceptibility mapping. *Catena*, *114*, 21-36.
- Pawluszek, K. & Borkowski, A. Nat Hazards (2017) 86: 919. doi:10.1007/s11069-016-2725-y
- Ahmed, B., & Dewan, A. (2017). Application of Bivariate and Multivariate Statistical Techniques in Landslide Susceptibility Modeling in Chittagong City Corporation, Bangladesh. Remote Sensing, 9(4), 304.

Additional drawbacks:

- Introduction

- nothing was said about machine learning methods to assess landslide susceptibility
- generally introduction section should be rewritten in order to give more flow.
- author started the introduction with describing the study area, the importance of the landslide hazard over there and the overview of the landslide susceptibility method without clear paragraph where the objective of the study is described (usually in the end of the introduction section)

- General situation of the study area
  - Based on the information provided in this section ("Weather condition along KKH are not uniform and are characterized by a wide range of annual mean temperatures and precipitation") it seems that study area cover a lot of square kilometers, this information is not provided in the manuscript.
  - I will encourage you to combine section 3 and 4 with the section 2 Study area and then create subsection general setting, geological setting, seismological setting etc.
- Subsection D- hydrology. Where is the image of this factor? What exactly has been used as hydrological factor? Proximity to the river? Precipitation? It is not clearly specified and it is not showed as a figure.
- I encourage you to create a table with all controlling parameter/ layers, which was used for the analysis with one column where the source of the data will be presented, number of classes, weights etc.
- Evaluation. It is written "According to the obtained results, most of the landslide events were found in high and very high susceptibility zone....." You didn't provide the number in the text and in table 4 there is no 4 susceptibility classes but 10 stability zones and landslide density over this zones. It is difficult to evaluate this map because of the heterogeneity.
- There is no information how many landslide was used to create the model, how many landslide to validate the model ? How man percentage?
- Line 9 "However, in our area these parameters seem to have a reduced influence on landslide occurrence" Based on what you are able to say this?
- English should be increased. For instance word geomorphologic or geomorphological is used interchangeably
- Weighted Overlay Method. Even when it is simple *"Weighted Sum"* tool in ArcGIS, some equations to this method are needed. Moreover, the references to this method are missing
- The rule how author classify the final weighed sum into 4 susceptibility zones is missing
- Figures:

Figure 6 Why spatial analysis of controlling factors have been made only for 6 layers not for all controlling parameters which were used? For instance, it will be good to see the how many landslide fall into the specific seismicity zone, land cover zone, hydrology etc. This part is missing.
Figure 4 - no scale
Figure 5 - it will be nice to see on this detailed geology overlaid with the KKH road
Figure 7 and 8 legend for  KKH road

Presented research should be extended emphasizing new aspects of the methodology,  providing calibration & validation of the proposed method, in addition to comparison with others available in literature. In the reviewer's opinion the manuscript et the present stage should be rejected. However, in case of very interesting study area (KKH) with diverse aspect influencing the landslide activity (seismicity, complex geology etc.) reviewer encourage authors to improve the methodology with some novel techniques (machine learning) or comparison between diverse techniques and resubmit this manuscript to NHESS.

---

## Author Comment (AC1) · 27 Apr 2018

**Response on Interactive comment by Anonymous Referee #1**

**Sajid Ali**

First, we would like to thank you for evaluation and highlighting the deficiencies in the manuscript. It is indeed valuable addition and help us to improve our manuscript. Please find below our response to Referee's comments.

**Comment:** First of all, the structure of this manuscript seems clear but some parts are missing or not well-balanced. Almost eight out of ten article pages, i.e. 80% of the written text deal with introduction, situation and methodology. The most important parts of a scientific article, results and discussion, fit in only one article page. In fact, the discussion chapter is missing at all, although some statements are "hidden" in the chapter "accuracy statement" and in the conclusions. This structure needs to be changed, and most important, a well-balanced discussion of the results has to elaborated. Please review your text passages thoroughly and sort primary information, results and discussion and clarify what the model outcomes mean scientifically.

**Response:** We will elaborate the results with a well-balanced discussion. We will add a case study to make it clearer.

Introduction:

**Comment:** I would appreciate more references on conditioning and triggering factors, and these can be used in the discussion again. Already in the first chapter, the insecurity of AHP is announced (p.2, line 29), and a combination with qualitative approaches and the use of GIS is considered a better option for regional studies.

**Response:** We will add more references on conditioning and triggering factors.

General situation of the study area:

**Comment:** How significant is the value of average annual precipitation in a Monsoon climate, also with abrupt changes to semi-arid / arid conditions?

**Response:** A strong association between rainfall intensity and mass movements along the Highway has been found in our previous publication (Ali et al. 2017). Rainfall is responsible for debris slides, debris flows and mud flows along the highway. The section of highway passing through semi-arid/arid zones has comparatively less number of landslides than that of passing through monsoon zone. However, Raikot Section lies in semi-arid/arid zone but contains multiple landslides because of close proximity with active Raikot Fault.

**Comment:** Are there more precise rainfall data available or can more datasets be taken into account in order to reduce the error source? Please clarify, how the precipitation data is used for the model.

**Response:** I have only one data source that is Pakistan Meteorological Department so I could not consider other data sources. I interpolated rainfall data of six weather stations to generate raster rainfall map.

Geology along the KKH:

**Comment:** "Highly active landslide zones were identified from the distribution of existing landslides" – what about other zones which may have not been activated yet? Actually you do not know only based on the distribution of visible deposits how active a distinct zone is at the moment or at what certain frequency and what type of trigger is the main causing factor of the landslides. There might be lots of "old" deposits summing up to a higher total number of landslides in a rather "inactive" zone compared to a recently activated zone with less, but young landslides. Please clarify how you mapped and characterized the landslides in the field. An attribute table of the landslides would be very helpful and support the discussion.

**Response:** For space management, we only showed the geology of four important sections. We considered detailed geology of each part of the Highway. For landslides inventory preparation, we considered data from road clearance logs of Frontier Works Organization and Geological Survey of Pakistan's publications. We have used record of landslides events of period 1982-82, 1996-2000 and 2014-16. Therefore, this landslide data gives an overview of 35 years.

Seismology:

**Comment:** The earthquake events in the region seem to be well-documented. How do you include these data in your hazard assessment? Especially the recent strong ones (e.g. the referenced in Oct 2005 or Oct 2015) could be used in a detailed case study. I would strongly recommend to do a detailed case study for at least one strong landslide event including lithology, seismology, conditioning and triggering factors. This would ideally fit your GIS-based mapping results. If not, this would also need to be discussed in detail.

**Response:** We used seismic intensities of the events (>6) for hazard assessment.

Thank you very much for your recommendation. We will include a case study in discussion chapter.

Geomorphologic factors:

**Comment:** How did you divide slope steepness into classes? I wonder on what criteria you based the data processing.

**Response:** Division of slope steepness into classes was based on statistical analysis. I tried different classes but found this division better in our study area.

Literature review:

**Comment:** Do you need this subsection? p.7, line 8: six weather stations along the highway: where are they (what climate?) and how does the rainfall influence your model? Please discuss, although (or because??) you classify the susceptibility levels based on active faults, seismic zones and steep slopes.

**Response:** We showed these weather stations with rainfall intensities in Fig. 1. We will make it visible in map and caption.

Field reconnaissance:

**Comment:** These data sets should be presented in more detail. A landslide inventory map with classification and indication of magnitude and/or frequency and a case study could improve this paper a lot. So far only the location (Fig. 1) and the number (72) of the mapped landslides are given. For instance, you could set up tables like the colleagues T. Stanley and D. Kirschbaum (2017) in their study on global landslide susceptibility mapping (Table 4 and Table 5).

**Response:** We will include landslide inventory map with types of failures. We think inventory map is better than table.

Remote sensing:

**Comment:** DEM-quality of 30x30 m2 and its influence on the results needs to be discussed, as several recent studies have shown that the DEM quality is crucial for modelling outcome. The accuracy of the land cover map of 87% needs to be discussed – is it a good or bad value compared to other studies?

**Response:** We will compare with previous land cover maps. We will discuss the limitations of data.

Weighted overlay method:

**Comment:** crucial part of this study. You should disclose the criteria.

In Table 2 the reader cannot follow your working procedure and how you chose the weighting factors.

**Response:** Firstly, we will explain Weighted overlay method more. Secondly, the procedure to weight the factors is based on pairwise comparisons to avoid biasness (Saaty 1987). It involves expert opinion and statistical analysis.

Results:

**Comment:** You need to explain the classification into four classes. What does a certain susceptibility level stand for? For this, table 3 could be developed in more detail (presence of active faults, seismic zones, steep slopes etc.).

Accuracy assessment: Please clarify, how the 72 landslide locations were used for validation. Only the location? What does the value of 72% tells us? Please discuss this outcome.

**Response:** We will explain classification and accuracy assessment.

**Comment:** Chapter discussion is missing!

**Response:** We will include a case study to make discussion chapter more understandable.

Conclusions:

**Comment:** Primary information, for instance the classification of landslides into rock-fall, debris slide etc., must be provided earlier in the results (landslide map, attribute table as suggested above). Furthermore, some interpretation is mixed in here (stable/quite stable parts of the highway). This interpretation and discussion is very important for the paper. You should lay out a new chapter, as indicated above, including limitations of analysis and sources of error.

**Response:** We will add an inventory map containing types of slope failures.

Table 4:

**Comment:** Please indicate the absolute values of area and number of landslides because this improves the transparency of your data processing. Why/ based on what criteria did you reduce the number of 9 susceptibility levels to 4 in your final map? Please clarify.

**Response:** We will indicate absolute values of area and observed landslides.

We converted 9 susceptibility levels into four equally with interval of two except *High Susceptibility* which contains susceptibility levels of 5, 6 and 7. We did so to distinguish the locations that are more hazardous.

Figure 1:

**Comment:** Please indicate subfigures a), b) and c) and legend. Too many (bold) lines in mid-zoom map.

**Response:** We will indicate and make lines light.

Figure 3:

**Comment:** Scale in W-E direction is missing.

**Response:** We will insert scale.

Figure 4b:

**Comment:** Why don't you also focus on the earthquake zone in the south of highlighted section a)? Not only lithology but also seismic situation should be considered here!

**Response:** We have considered seismic intensity of these events along with active faults (MMT) and lithology.

Figure 5:

**Comment:** Where is the highway? Please improve visibility.

**Response:** We will improve visibility.

Figure 6:

**Comment:** a) How would your model change, if you split the group of 31-45° again? Why do you show the group >65° and in subfigure d) the group 4000-4700m? Is the y-axis "landslide %" based on all 72 landslides? Please clarify. Please use these values in the suggested landslide map/attribute table (case study) and compare with other studies.

**Response:** We tried the interval of 10° but results were random. We showed it because the elevation of the Highway is 4604 meters. No, this landslide% is not based on 72 landslides but on all landslides events of period 1982-82, 1996-2000 and 2014-16. We only used random 72 landslides from this data set for validation purpose.

Figure 10/11:

**Comment:** Captions of subfigures a), b) and c) are missing. One of these recent events might be suitable for a detailed case study.

**Response:** We will insert above-mentioned captions.

---

## Author Comment (AC2) · 27 Apr 2018

**Sajid Ali**

First, we would like to thank you for evaluation and highlighting the deficiencies in the manuscript. It is indeed valuable addition and help us to improve our manuscript. Please find below our response to Referee's comments.

**Comment:** Generally, the issue considered in the manuscript is very important to Earth sciences community and could be publish in Earth System Sciences. However, the manuscript does not have the high scientific level presented by the journal.

The issue considered in the manuscript has been deeply discussed for decades, with many papers, methodologies and case studies. New works are more than welcome as long as they contribute something new to Earth science community, which may be new approaches, new case studies, and of course an improvement over the methods already published. Unfortunately this paper, even if engaging, doesn't offer anything new.

Here goes the list of critical shortcomings:

Generally, WOL method presented in this study was firstly presented in 1994 in work:

Bonham-Carter, G.F., 1994. Geographic information systems for geoscientists: modelling with GIS. Pegamon Press, Oxford.

Since this time a lot of improvements, strategies, combination, comparison with other techniques have been presented in scientific community, the method is not new. Please check this papers with novelties in similar methodologies:

- Yalcin, A. (2008). GIS-based landslide susceptibility mapping using analytical hierarchy process and bivariate statistics in Ardesen (Turkey): comparisons of results and confirmations. Catena, 72(1), 1-12
- Althuwaynee, O. F., Pradhan, B., Park, H. J., & Lee, J. H. (2014). A novel ensemble bivariate statistical evidential belief function with knowledge based analytical hierarchy process and multivariate statistical logistic regression for landslide susceptibility mapping. Catena, 114, 21-36.
- Pawluszek, K. & Borkowski, A. Nat Hazards (2017) 86: 919. doi:10.1007/s11069-016-2725-y
- Ahmed, B., & Dewan, A. (2017). Application of Bivariate and Multivariate Statistical Techniques in Landslide Susceptibility Modeling in Chittagong City Corporation, Bangladesh. Remote Sensing, 9(4), 304.

**Response:** The Karakoram Highway is an important physical connection between Pakistan and China and is the talk of the town these days due to its inclusion in China Pakistan Economic Corridor.

This contribution on above subject along this particular highway is first of its kind. In addition to it, diverse geology, seismology, tectonics, climate and geomorphology along the highway makes it unique.

We used landslide data of different periods (1982-82, 1996-2000, 2014-16) and then considered almost all aspects (Geomorphology, Geology, Seismology, Weather), which can contribute to slope failures.

Lastly, as recommended by first Referee, we will include a case study in our discussion chapter.

According to us, it will be an important contribution to this special issue "Landslide–transport network interactions".

**Additional drawbacks:**

**Introduction**

**Comment:** nothing was said about machine learning methods to assess landslide susceptibility

**Response:** We are not using any machine learning techniques.

**Comment:** generally introduction section should be rewritten in order to give more flow. - author started the introduction with describing the study area, the importance of the landslide hazard over there and the overview of the landslide susceptibility method without clear paragraph where the objective of the study is described (usually in the end of the introduction section).

**Response:** We will include objectives and more references.

**General situation of the study area**

**Comment:** Based on the information provided in this section ("Weather condition along KKH are not uniform and are characterized by a wide range of annual mean temperatures and precipitation") it seems that study area cover a lot of square kilometers, this information is not provided in the manuscript.

**Response:** We already mentioned its 840 km long highway, but we can also mention area of the susceptibility map.

**Comment:** I will encourage you to combine section 3 and 4 with the section 2 Study area and then create subsection general setting, geological setting, seismological setting etc.

**Response:** We have made these sections to facilitate the reader.

**Comment:** **Subsection D- hydrology**. Where is the image of this factor? What exactly has been used as hydrological factor? Proximity to the river? Precipitation? It is not clearly specified and it is not showed as a figure.

**Response:** We have used rainfall intensity. Figure 4 is showing different weather conditions along the highway.

**Comment:** I encourage you to create a table with all controlling parameter/ layers, which was used for the analysis with one column where the source of the data will be presented, number of classes, weights etc.

**Response:** We will include this.

**Comment:** Evaluation. It is written "According to the obtained results, most of the landslide events were found in high and very high susceptibility zone….." You didn't provide the number in the text and in table 4 there is no 4 susceptibility classes but 10 stability zones and landslide density over this zones. It is difficult to evaluate this map because of the heterogeneity.

**Response:** We will indicate absolute values of area and observed landslides as proposed by first Referee.

**Comment:** There is no information how many landslide was used to create the model, how many landslide to validate the model ? How man percentage?

**Response:** We used 72 random landslides from whole inventory to validate the map. The addition of a separate inventory map indicating all slope failures (rock-fall, debris slide) will resolve the problem.

**Comment:** Line 9 "However, in our area these parameters seem to have a reduced influence on landslide occurrence" Based on what you are able to say this?

**Response:** Based on statistical analysis, we were able to say.

**Comment:** English should be increased. For instance word geomorphologic or geomorphological is used interchangeably

**Response:** We will request our colleague who is native English speaker for proof reading.

**Comment:** Weighted Overlay Method. Even when it is simple "*Weighted Sum"* tool in ArcGIS, some equations to this method are needed. Moreover, the references to this method are missing

**Response:** We will add references.

**Comment:** The rule how author classify the final weighed sum into 4 susceptibility zones is missing.

**Response:** We will explain it.

**Figures:**

**Comment:** Figure 6 Why spatial analysis of controlling factors have been made only for 6 layers not for all controlling parameters which were used? For instance, it will be good to see the how many landslide fall into the specific seismicity zone, land cover zone, hydrology etc. This part is missing.

**Response:** We will add it.

**Comment:** Figure 4 - no scale

**Response:** We will insert it.

**Comment:** Figure 5 - it will be nice to see on this detailed geology overlaid with the KKH road
**Response:** We will overlay the highway.

**Comment:** Figure 7 and 8 legend for KKH road.
**Response:** We will add legend for KKH.

---

## Referee Comment (RC3) · Anonymous Referee #3 · 10 May 2018

Authors use the well known AHP method to obtain a map of landslide susceptibility 'along' the Karakoram Highway. Explanatory variables are prepared and classified with a variety of different approaches. The topic is relevant and for sure a correct map assessing the susceptibility along the highway would/might help decision makers to mitigate the risk. Unfortunately there are several issues about the work done and the way it is presented that do not make it, in my opinion, suitable for publication.

In general:

the method is already well known in literature and the paper does not seem to me to introduce any originality, it's more like a technical report where a consolidated procedure

is used to obtain a map.

The description of the input data is very wanting, in particular the main information related to landslides is practically non existent, something very strange if a landslide susceptibility map is to be prepared. This aspect somehow also invalidates the validation of the map.

Most of the decisions taken in the choice of the model and in its setup are not justified but just given. The literature analysis does not help at all.

Some of the important details are not provided and only in few cases it is possible to deduce or extrapolate the needed information. An example is the study area: is it 2 km large? It's just a 'rigid' buffer along the highway or it is obtained using geomorphological analysis (slopes, catchments?). According a figure, it should be a 'strip' some km large, but it's my deduction, if true, why this choice?

I was expecting (or hoping) for a discussion to find reasons for the choices, analysis in deep of the consequences of the choices on the final results, but this part is missing.

My personal opinion is that the paper does not reach the minimum standards of quality from too many points of view and it is to be rejected, perhaps deeply re-thought, and eventually resubmitted.

My considerations are, I hope, supported by my comments in detail shown below

Abstract:

Abstract is a copy and paste of some parts of the paper

p1 l14: not sure you need to go so much into geological details in the abstract, just say that geology facilitates the landslide occurrence.

p1 l23: something unclear here.

p1 l27: validation of the credibility?

1 Introduction

p1 l31 - p2 l3: these first 5 lines should be moved to the area description paragraph

p2 l5: why potentially?

p2 l7: no, landslides are not caused by conditioning factors, they might eventually facilitate the occurrence.

p2 l9: too generic, variation of geology? Geological structure?

p2 l10: too generic... how facilitated?

General comment: the literature analysis is sterile because it does not motivate/support any of the choices taken in the procedure setup.

2. General situation of the study area

I actually cannot understand how is the shape of the study area, is it a buffer (what kind of buffer) along the KKT? How large, how is it defined?

3 Geology along the KKH

p3 l29: active landslide zones: what does that mean? and how can you get it from the distribution of existing landslides? I guess you mean that the susceptibility in the area is high...

p4 l1: again I'm not sure that 'activity' is here used correctly, I suggest to re-phrase to avoid confusion with activity of a landslide which is another thing.

p4 l17: see my previous comment.

4 Seismology

5 Causative factors and spatial distribution analysis

p5 l14: all those parameters are always present, you probably mean depending on the values, or classes...

p5 l15: what do you mean here with accurate and precise?

p5 l15: entirely dependent on the availability of data relating to controlling factors? Not sure this is in general true, and in particular here using this type of model.

a Lithology

p5 l19: what do you mean with time?

p5 l21: what do you mean with spatial analysis? Did you count landslides for each lithology, or it is a spatial density (landslide area / area)?

b distance from faults

c Geomorphological factors

p6 l5 - p6 l9: the result of the numerical distribution of the landslides inside the classes might depend (actually it will depend for sure) on how the classes were chosen that is not described.

p6 l6 -p6 l9: what do you mean with reduced? According to fig. 6 they seem to have some discriminating capabilities, I think you should support more your conclusion (perhaps correct, but should not be deduced just by looking at fig. 6)

d Hydrology

p6 l15: no doubts about the correlation and the work done by Ali et al., but her it should be better introduced and contextualized.

e Land cover

This is more a description of the land cover of the area instead of a spatial analysis that is entrusted to the reader.

6 Methodology

a Literature review

[Figure]

The description of the data here is really poor and unsatisfactory. In particular, the description on how the inventories are and they were prepared is completely missing. How many landslides, type, some statistical description... landslides are here the secondary variable, the one that is used to understand how and if the causing factors are (quantitatively) important or not. Furthermore, what is a precipitation map? Is this an annual precipitation map? How did you obtain it? How to make sure that it is not too much event dependent (the bias introduced in the model would be dramatic)..

b Field reconnaissance

p7 l11: for all landslides?

p7 l12 - p11 l13: what does that mean? Another inventory? Is this a buffer around the highway where the map was prepared?

c Remote sensing

Same impression I had about 'Literature review'.

What is the level of pre-processing of the satellite images (are they orthorectified? Atmospheric corrected?), what did you do with QGIS. How did you train the classification, how many ROIs, what model did you use (SVM, ML...)? The map is not divided in 4 classes, probably you did look for 4 classes in the training phase... The confusion method of the map versus what?? The image previously classified? How, by who, so whi did not use it?

d Analytical hierarchy process

I think you should refer the explanation to table 2. (and not only in the next paragraph). Not sure you can say x,y axes. I also suggest you to add some references to find easily what CR and CI are.

7 Results.

p8 l15 - p8 l16: how did you choose the classes?

p8 l17: these numbers depend on the previous classification that was not justified, so they say nothing.

p8 l20: what kind of criterion is an 'owed to lucidity'?

a Accuracy assessment

General question: did you use all landslides in the inventory? A part of? I can't understand what is the classifying parameter.

p9 l8: among many you pick up ROC and LDA without justification.

8 Conclusions

p9 l23: a set?
* * *

---

## Author Comment (AC3) · 18 May 2018

**Sajid Ali**

First, we would like to thank you for evaluation and highlighting the deficiencies in the manuscript. It is indeed valuable addition and help us to improve our manuscript. Please find below our response to Referee's comments.

**Comment:** The method is already well known in literature and the paper does not seem to me to introduce any originality, it's more like a technical report where a consolidated procedure is used to obtain a map.

The description of the input data is very wanting, in particular the main information related to landslides is practically non existent, something very strange if a landslide susceptibility map is to be prepared. This aspect somehow also invalidates the validation of the map.

Most of the decisions taken in the choice of the model and in its setup are not justified but just given. The literature analysis does not help at all. Some of the important details are not provided and only in few cases it is possible to deduce or extrapolate the needed information. An example is the study area: is it 2 km large? It's just a 'rigid' buffer along the highway or it is obtained using geomorphological analysis (slopes, catchments?). According a figure, it should be a 'strip' some km large, but it's my deduction, if true, why this choice?

I was expecting (or hoping) for a discussion to find reasons for the choices, analysis in deep of the consequences of the choices on the final results, but this part is missing. My personal opinion is that the paper does not reach the minimum standards of quality from too many points of view and it is to be rejected, perhaps deeply re-thought, and eventually resubmitted.

**Response:** We explained the choice of the model on Page2|20-33 and will explain it more. Furthermore, the addition of valuable suggestion/comments from all three Referees would improve the data presentation and quality of the paper. In addition, the inclusion of a case study in discussion part will make it a valuable addition to this special issue.

*My considerations are, I hope, supported by my comments in detail shown below:*

Abstract:
**Comment:** Abstract is a copy and paste of some parts of the paper
**Response:** We tried to give an overview of the publication.

**Comment:** p1 l14: not sure you need to go so much into geological details in the abstract, just say that geology facilitates the landslide occurrence.
**Response:** Ok! We will do it. However, this is exactly the point in your comments to explain the input data.

**Comment:** p1 l27: validation of the credibility?
**Response:** the soundness and reliability was demonstrated by accuracy assessment.

1 Introduction

**Comment:** p1 l31 - p2 l3: these first 5 lines should be moved to the area description paragraph
**Response:** Ok we will do it.

**Comment:** p2 l5: why potentially?
**Response:** It is "momentous". Therefore, we will remove "potentially".

**Comment:** p2 l7: no, landslides are not caused by conditioning factors, they might eventually facilitate the occurrence.
**Response:** Without facilitation or pre-conditioning of the rockmass, landslides would not occur. For example, Seismicity and rainfall easily trigger those slopes, which have already been pre-conditioned means highly fractured.

**Comment:** p2 l9: too generic, variation of geology? Geological structure?
**Response:** means both conditioning and triggering factors!

**Comment:** p2 l10: too generic... how facilitated?
**Response:** We will explain it more!

**Comment:** General comment: the literature analysis is sterile because it does not motivate/support any of the choices taken in the procedure setup.
**Response:** We will explain it more!

2. General situation of the study area

**Comment:** I actually cannot understand how is the shape of the study area, is it a buffer (what kind of buffer) along the KKT? How large, how is it defined?
**Response:** It is buffer of 5 km along the Highway.

3 Geology along the KKH

**Comment:** p3 l29: active landslide zones: what does that mean? and how can you get it from the distribution of existing landslides? I guess you mean that the susceptibility in the area is high...
**Response:** It is referring back to Fig. 1, where an inventory of landslides of three different periods (1982-82, 1996-2000, 2014-16) is shown. Because of multiple episodes of slope failure within these zones, we termed it as active landslide zones.

**Comment:** p4 l1: again I'm not sure that 'activity' is here used correctly, I suggest to re-phrase to avoid confusion with activity of a landslide which is another thing.
**Response:** We will re-phrase it.

**Comment:** p4 l17: see my previous comment.
**Response:** Due to multiple episodes of slope failure in different periods (1982-82, 1996-2000, 2014-16), we termed it as active landslide zone.

4 Seismology

5 Causative factors and spatial distribution analysis

**Comment:** p5 l14: all those parameters are always present, you probably mean depending on the values, or classes...
**Response:** Yes, you are right! we will re-phrase it.

**Comment:** p5 l15: what do you mean here with accurate and precise?
**Response:** Accuracy is the closeness of the information on a susceptibility map with the real values whereas precision is the exactness of the description of data.

**Comment:** p5 l15: entirely dependent on the availability of data relating to controlling factors? Not sure this is in general true, and in particular here using this type of model.
**Response:** The resolution of DEM and satellite images and the scale of the geological maps have a direct impact on the quality of susceptibility map.

a Lithology

**Comment:** p5 l19: what do you mean with time?
**Response:** Mechanical behavior of some formations changes quickly even after very low rainfall and fails whereas some formations take a bit longer to reach at that failure stage. So time of slope failure also depends on lithology.

**Comment:** p5 l21: what do you mean with spatial analysis? Did you count landslides for each lithology, or it is a spatial density (landslide area / area)?
**Response:** It is spatial density analysis.

b distance from faults

c Geomorphological factors

**Comment:** p6 l5 - p6 l9: the result of the numerical distribution of the landslides inside the classes might depend (actually it will depend for sure) on how the classes were chosen that is not described.
**Response:** We will explain it more!

**Comment:** p6 l6 -p6 l9: what do you mean with reduced? According to fig. 6 they seem to have some discriminating capabilities, I think you should support more your conclusion (perhaps correct, but should not be deduced just by looking at fig. 6)
**Response:** We will explain it more!

d Hydrology

**Comment:** p6 l15: no doubts about the correlation and the work done by Ali et al., but her it should be better introduced and contextualized.
**Response:** We will explain it more!

e Land cover

**Comment:** This is more a description of the land cover of the area instead of a spatial analysis that is entrusted to the reader.
**Response:** We will also discuss spatial analysis.

6 Methodology

a Literature review

**Comment:** The description of the data here is really poor and unsatisfactory. In particular, the description on how the inventories are and they were prepared is completely missing. How many landslides, type, some statistical description... landslides are here the secondary variable, the one that is used to understand how and if the causing factors are (quantitatively) important or not. Furthermore, what is a precipitation map? Is this an annual precipitation map? How did you obtain it? How to make sure that it is not too much event dependent (the bias introduced in the model would be dramatic).
**Response:** We will explain more and include landslide inventory map with types of failures as advised by first Referee. It is annual precipitation map. We interpolated rainfall data of six weather stations to generate rainfall map. We used spatial analysis results and Analytical Hierarchy Process (AHP) involving expert opinion to rate all parameters. It has reduced the chances of possible bias.

b Field reconnaissance

**Comment:** p7 l11: for all landslides?
**Response:** Yes! We did for all landslides.

**Comment:** p7 l12 - p11 l13: what does that mean? Another inventory? Is this a buffer around the highway where the map was prepared?
**Response:** Landslides within area of $2km^2$ around the highway were only considered during preparation of Inventory.

c Remote sensing

**Comment:** Same impression I had about 'Literature review'.
**Response:** We will explain more!

**Comment:** What is the level of pre-processing of the satellite images (are they orthorectified? Atmospheric corrected?), what did you do with QGIS. How did you train the classification, how many ROIs, what model did you use (SVM, ML...)? The map is not divided in 4 classes, probably you did look for 4 classes in the training phase... The confusion method of the map versus what?? The image previously classified? How, by who, so whi did not use it?
**Response:** These images were pre-processed on QGIS. We used Maximum Likelihood Classification on Arc GIS 10.3. Yes! We looked for four classes in the training phase. We will explain this part more.

d Analytical hierarchy process

**Comment:** I think you should refer the explanation to table 2. (and not only in the next paragraph). Not sure you can say x,y axes. I also suggest you to add some references to find easily what CR and CI are.
**Response:** We will add more references!

7 Results.

**Comment:** p8 l15 - p8 l16: how did you choose the classes?
**Response:** We converted nine (9) susceptibility levels into four equally with interval of two except High Susceptibility which contains susceptibility levels of 5, 6 and 7. We did so to distinguish the locations that are more hazardous.

**Comment:** p8 l17: these numbers depend on the previous classification that was not justified, so they say nothing.
**Response:** We will explain that!

**Comment:** p8 l20: what kind of criterion is an 'owed to lucidity'?
**Response:** By keeping in mind the scale of the map, the visibility of a single map of 840 km long highway was an issue. So make it clearer to readers, we divided into two parts.

a Accuracy assessment

**Comment:** General question: did you use all landslides in the inventory? A part of? I can't understand what is the classifying parameter.
**Response:** We randomly selected 72 landslides from the inventory to validate the map.

**Comment:** p9 l8: among many you pick up ROC and LDA without justification.
**Response:** We will explain it more!

8 Conclusions

**Comment:** p9 l23: a set?
**Response:** means two maps. Thank you very much for pointing it out! We will rephrase it, as it is a single map!

---

## Author Response (AR1)

| COMMENT                                                                                                                                                                                                                                                                                                                                                                                                                                                                                                                                                                                                                                                                                                                                                                                                                  | RESPONSE                                                                                                                                                                 |  |  |  |  |
|--------------------------------------------------------------------------------------------------------------------------------------------------------------------------------------------------------------------------------------------------------------------------------------------------------------------------------------------------------------------------------------------------------------------------------------------------------------------------------------------------------------------------------------------------------------------------------------------------------------------------------------------------------------------------------------------------------------------------------------------------------------------------------------------------------------------------|--------------------------------------------------------------------------------------------------------------------------------------------------------------------------|--|--|--|--|
| Anonymous                                                                                                                                                                                                                                                                                                                                                                                                                                                                                                                                                                                                                                                                                                                                                                                                                | Referee #1                                                                                                                                                               |  |  |  |  |
| First of all, the structure of this manuscript seems
clear but some parts are missing or not well-balanced.
Almost eight out of ten article pages, i.e. 80% of the
written text deal with introduction, situation and
methodology. The most important parts of a scientific
article, results and discussion, fit in only one article
page. In fact, the discussion chapter is missing at all,
although some statements are "hidden" in the
chapter "accuracy statement" and in the conclusions.
This structure needs to be changed, and most
important, a well-balanced discussion of the results
has to elaborated. Please review your text passages
thoroughly and sort primary information, results and
discussion and clarify what the model outcomes mean
scientifically. | We added three case studies to elaborate the results
(p29 – p34).                                                                                                     |  |  |  |  |
| I would appreciate more references on conditioning
and triggering factors, and these can be used in the
discussion again. Already in the first chapter, the
insecurity of AHP is announced (p.2, line 29), and a
combination with qualitative approaches and the use
of GIS is considered a better option for regional
studies.                                                                                                                                                                                                                                                                                                                                                                                                                                                                        | We added more references on conditioning and triggering factors (p2 2 – p2 16).                                                                                          |  |  |  |  |
| Do you need this subsection? p.7, line 8: six weather
stations along the highway: where are they (what
climate?) and how does the rainfall influence your
model? Please discuss, although (or because??) you
classify the susceptibility levels based on active faults,
seismic zones and steep slopes.                                                                                                                                                                                                                                                                                                                                                                                                                                                                                                   | We made Fig. 1 more clear and visible (p4).                                                                                                                              |  |  |  |  |
| These data sets should be presented in more detail. A landslide inventory map with classification and indication of magnitude and/or frequency and a case study could improve this paper a lot. So far only the location (Fig. 1) and the number (72) of the mapped landslides are given. For instance, you could set up tables like the colleagues T. Stanley and D. Kirschbaum (2017) in their study on global landslide susceptibility mapping (Table 4 and Table 5).                                                                                                                                                                                                                                                                                                                                                 | We included multi-temporal landslide inventory map
with types of failures and a table. We added a full
subsection about landslide inventory map (p17 10 –
p20). |  |  |  |  |
| DEM-quality of 30x30 m2 and its influence on the results needs to be discussed, as several recent studies have shown that the DEM quality is crucial for modelling outcome. The accuracy of the land cover map of 87% needs to be discussed – is it a good or bad value compared to other studies?                                                                                                                                                                                                                                                                                                                                                                                                                                                                                                                       | We added data based limitations at the end of
conclusion (p37 5 – 8)
We also compared land cover map with previous
maps (p14 2 – p14 12)                        |  |  |  |  |

| Chapter discussion is missing!                                                                                                                                                                                                                                                                                                                                                                                                                                                                             | We added case studies as advised.                                                                  |
|------------------------------------------------------------------------------------------------------------------------------------------------------------------------------------------------------------------------------------------------------------------------------------------------------------------------------------------------------------------------------------------------------------------------------------------------------------------------------------------------------------|----------------------------------------------------------------------------------------------------|
| Primary information, for instance the classification of
landslides into rock-fall, debris slide etc., must be
provided earlier in the results (landslide map,
attribute table as suggested above). Furthermore,
some interpretation is mixed in here (stable/quite
stable parts of the highway). This interpretation and
discussion is very important for the paper. You should
lay out a new chapter, as indicated above, including
limitations of analysis and sources of error. | We added landslide inventory map and table (p17 10
– p20).                                      |
| Please indicate the absolute values of area and
number of landslides because this improves the
transparency of your data processing. Why/ based on
what criteria did you reduce the number of 9
susceptibility levels to 4 in your final map? Please
clarify.                                                                                                                                                                                                                               | We indicated absolute value in the same table (p35).
We explained it (p24 16-18)                |
| Please indicate subfigures a), b) and c) and legend.
Too many (bold) lines in mid-zoom map.                                                                                                                                                                                                                                                                                                                                                                                                             | We corrected as advised (p4)                                                                       |
| Figure 3: Scale in W-E direction is missing.                                                                                                                                                                                                                                                                                                                                                                                                                                                               | We added scale (p6)                                                                                |
| Figure 5: Where is the highway? Please improve visibility.                                                                                                                                                                                                                                                                                                                                                                                                                                                 | We improved visibility of the KKH (p9)                                                             |
| Figure 10/11: Captions of subfigures a), b) and c) are missing. One of these recent events might be suitable for a detailed case study.                                                                                                                                                                                                                                                                                                                                                                    | We inserted above-mentioned captions (p26 – p27)                                                   |
| Anonymous                                                                                                                                                                                                                                                                                                                                                                                                                                                                                                  | s Referee #2                                                                                       |
| Generally, introduction section should be rewritten in
order to give more flow author started the
introduction with describing the study area, the
importance of the landslide hazard over there and the
overview of the landslide susceptibility method
without clear paragraph where the objective of the
study is described (usually in the end of the
introduction section).                                                                                                      | We added more references (p2 2-19) & (p3 3-14)                                                     |
| I encourage you to create a table with all controlling
parameter/ layers, which was used for the analysis
with one column where the source of the data will be
presented, number of classes, weights etc.                                                                                                                                                                                                                                                                                         | We added table (p23)                                                                               |
| Evaluation. It is written "According to the obtained
results, most of the landslide events were found in
high and very high susceptibility zone" You didn't
provide the number in the text and in table 4 there is
no 4 susceptibility classes but 10 stability zones and
landslide density over this zones. It is difficult to                                                                                                                                                             | We indicated absolute values of area and observed
landslides as proposed by first Referee (p35) |

| evaluate this map because of the heterogeneity.                                                                                                                                                                                                                                                                    |                                                                                                                                                                             |  |  |  |  |
|--------------------------------------------------------------------------------------------------------------------------------------------------------------------------------------------------------------------------------------------------------------------------------------------------------------------|-----------------------------------------------------------------------------------------------------------------------------------------------------------------------------|--|--|--|--|
|                                                                                                                                                                                                                                                                                                                    |                                                                                                                                                                             |  |  |  |  |
| The rule how author classify the final weighed sum into 4 susceptibility zones is missing.                                                                                                                                                                                                                         | We explained it (p25 3-5).                                                                                                                                                  |  |  |  |  |
| Figure 6 Why spatial analysis of controlling factors
have been made only for 6 layers not for all
controlling parameters which were used? For
instance, it will be good to see the how many
landslide fall into the specific seismicity zone, land
cover zone, hydrology etc. This part is missing. | As it has become very long paper due to many figures.
Therefore, We didn't added.
But spatial analysis results for landcover and rainfall
are integrated (p6,p15). |  |  |  |  |
| Figure 4 - no scale                                                                                                                                                                                                                                                                                                | Scale is added (p8)                                                                                                                                                         |  |  |  |  |
| Figure 5 - it will be nice to see on this detailed geology overlaid with the KKH road                                                                                                                                                                                                                              | We overlaid the KKH (p9)                                                                                                                                                    |  |  |  |  |
| Figure 7 and 8 legend for KKH road.                                                                                                                                                                                                                                                                                | We added legend for KKH (p12 & p15)                                                                                                                                         |  |  |  |  |
| Anonymous                                                                                                                                                                                                                                                                                                          | s Referee #3                                                                                                                                                                |  |  |  |  |
| p1 l31 - p2 l3: these first 5 lines should be moved to the area description paragraph                                                                                                                                                                                                                              | We moved these lines to description paragraph (p3 20-26).                                                                                                                   |  |  |  |  |
| p2 l5: why potentially?                                                                                                                                                                                                                                                                                            | We removed word potentially.                                                                                                                                                |  |  |  |  |
| p2 l10: too generic how facilitated?                                                                                                                                                                                                                                                                               | We explained (P2 18-19)                                                                                                                                                     |  |  |  |  |
| General comment: the literature analysis is sterile
because it does not motivate/support any of the
choices taken in the procedure setup.                                                                                                                                                                    | We explained it (p3 3-14)                                                                                                                                                   |  |  |  |  |
| p4 11: again 1'm not sure that 'activity' is here used correctly, I suggest to re-phrase to avoid confusion with activity of a landslide which is another thing.                                                                                                                                                   | We rephrased it (p7 9-10).                                                                                                                                                  |  |  |  |  |
| p5 114: all those parameters are always present, you probably mean depending on the values, or classes                                                                                                                                                                                                             | We rephrased it (p10 19).                                                                                                                                                   |  |  |  |  |
| Geomorphological factors: p6 I5 - p6 I9: the result of
the numerical distribution of the landslides inside the
classes might depend (actually it will depend for sure)
on how the classes were chosen that is not described.                                                                              | We explained it (p13 5-6).                                                                                                                                                  |  |  |  |  |
| p6 I15: no doubts about the correlation and the work
done by Ali et al., but her it should be better
introduced and contextualized.                                                                                                                                                                          | We explained it more (p13 16-23).                                                                                                                                           |  |  |  |  |
| This is more a description of the land cover of the area
instead of a spatial analysis that is entrusted to the
reader.                                                                                                                                                                                      | We included spatial analysis (p14 16-20)                                                                                                                                    |  |  |  |  |

| The description of the data here is really poor and     | We included multi-temporal landslide inventory map  |
|---------------------------------------------------------|-----------------------------------------------------|
| unsatisfactory. In particular, the description on how   | with types of failures and a table. We added a full |
| the inventories are and they were prepared is           | subsection about landslide inventory map (p17 10 –  |
| completely missing. How many landslides, type, some     | p20).                                               |
| statistical description landslides are here the         |                                                     |
| secondary variable, the one that is used to understand  |                                                     |
| how and if the causing factors are (quantitatively)     | We explained it (p13 16-23) & (p17 8-9).            |
| important or not. Furthermore, what is a precipitation  |                                                     |
| map? Is this an annual precipitation map? How did       |                                                     |
| you obtain it? How to make sure that it is not too      |                                                     |
| much event dependent (the bias introduced in the        |                                                     |
| model would be dramatic).                               |                                                     |
| Remote sensing: Same impression I had about             | We explained it more (p21 11-18)                    |
| 'Literature review'.                                    |                                                     |
| What is the level of pre-processing of the satellite    | We explained it (p14 2-12)                          |
| images (are they orthorectified? Atmospheric            |                                                     |
| corrected?), what did you do with QGIS. How did you     |                                                     |
| train the classification, how many ROIs, what model     |                                                     |
| did you use (SVM, ML)? The map is not divided in 4      |                                                     |
| classes, probably you did look for 4 classes in the     |                                                     |
| training phase The confusion method of the map          |                                                     |
| versus what?? The image previously classified? How,     |                                                     |
| by who, so whi did not use it?                          |                                                     |
| Analytical hierarchy process: I think you should refer  | We explained it more references (p22 3 - p24 8)     |
| the explanation to table 2. (and not only in the next   |                                                     |
| paragraph). Not sure you can say x,y axes. I also       |                                                     |
| suggest you to add some references to find easily       |                                                     |
| what CR and CI are.                                     |                                                     |
| Results: p8 l17: these numbers depend on the            | We explained (p25 3-5)                              |
| previous classification that was not justified, so they |                                                     |
| say nothing.                                            |                                                     |
| Accuracy assessment: p9 l8: among many you pick up      | We justified it (p35 4-7)                           |
| ROC and LDA without justification.                      |                                                     |

[revised manuscript text omitted]

---

## Referee Report (RR1)

**Comments on**

**"Landslide susceptibility mapping by using GIS along the China Pakistan economic corridor (Karakoram Highway), Pakistan"**

- It should be noted that the idea of landslide susceptibility mapping using AHP method is not novel. It was largely explored by various researchers. For me, it is a case study and studies are classified as a "Technical note or report". *The readers have high expectation from a high-quality journal such as Nat. Hazards Earth Syst. Sci. by high Impact Factor (IF).* Although I appreciate the effort of authors to develop regional landslide susceptibility map based on the field data pertinent to actual landslides.

- Equally, the AHP is not without problems. Weights are derived in the AHP procedure but these weights themselves are dependent on the a priori classification, which can be suboptimal. In the present manuscript, it is slightly disappointing to see improvements that cannot be explained or traced back to data interaction or model errors.

- There is a real lack of geomorphological expertise in the relationship between variables and landslides. You accumulate numbers from statistics but the reader does not see what the actual contribution of your statistics, what it brings in addition to the knowledge of the phenomena.

- The paper has a lack of flow. I have some problems with figures that are poorly explained or difficult to follow.

- Only the application of the model is not a big deal, *researchers should have responsibility and accountability with their results*. What are the lessons to be remembered? For example, are there characteristic portions of the landscape that can be identified that are landslide prone?

**General comments**

1) Abstract
    a) The abstract seems copy paste of some sentences forms the main body of the manuscript. Not need to describe how Himalaya was formed in the manuscript. Highlight what is existing in instability problem and what is your contribution.

2) Introduction
    a) The introduction provides somehow sufficient background information, however, the authors need to provide an explanation about the necessity using the proposed models within the research in the particular area.
    b) "Some geoscientist incorporate………" you have to put more consistent references.

c) In the introduction, the objectives and hypothesis you be established and findings shall be written in your result section. How your research is different from others and what is your specific idea? should be mentioned.

3) General situation of the study area

   a) I am confused with your study area, is it just a buffer area of the road, right? But after reading this section and geology part, it seems more about regional scale containing unnecessary areas.

   b) The geological description is out of the scope, it's like a report. You should describe the effect of geological units and structure and existence of landslide.

4) Causative Factors and Spatial Distribution analysis

   a) How and why did you select 10 conditioning factors? What is the basis of that?

   b) It is not clear how selected conditioning factors influence generation of landslides?

   c) In this large area, only major faults play an important role? Included lineament attributes.

   d) How did you make classes for continuous data such as slope, elevation, drainage proximity, curvature etc.? Do have you try to make a sensitivity analysis of different classes? If you aggregate the three first class, the final probabilities increase in these locations? You do not make sensitivity analysis on the different data and classes, but it is the first step to have a robust strategy of LSA and robust final susceptibility map!

   e) How did you prepare the rainfall map? Is that annual max rainfall, mean rainfall, event base rainfall or total rainfall etc?

   f) Is the proximity of streams only one hydrological factor? You have DEM, why did not you generate others like TWI, SPI, STI etc?

   g) Explain how changes in landcover control the spatial distribution of landslide in your study area. Did you compare imageries?

5) Methodology

   a) A poor architect was presented in the methodological part.

   b) What is the real improvement of the field reconnaissance survey? How many landslides have been corrected in term of locations and type? You have to put more details about the approach, results and improvements. You can take example about that in the parer of Fressard et al., 2013. They compared different inventories and the improvements of field survey. Please explain why you focus on thes landslide types...

   c) Give reference to SCP. It is not clear the accuracy assessment of landcover map. This should be in result part not in the methodological section.

6) Results
   a) With threshold do you used to split the weighted map into four categories? How do you choose these values? This is an important point that is not justified or described in the text.

**No discussion paragraph?** The authors estimate that their approach does not require a minimum of objective criticism? This is very pretentious because there is much to be said about the strategy of calibration/validation of study approach.

**Remark:**

As it was written earlier in the comments: LSA requires a real expertise and knowledge of field to make susceptibility maps. It is not enough to make statistics and put themin an article to make this article a scientific publication !

---

## Author Response (AR2)

First, we would like to thank you for evaluation and highlighting the deficiencies in the manuscript. It is indeed valuable addition and help us to improve our manuscript. Please find below our response to Editor, Anonymous Referee #4 and Anonymous Referee #5's comments.

| COMMENT | RESPONSE |
|---|---|
| **Editor Comments** | |
| Towards the beginning, a review of any previous work addressing landslide susceptibility in the Karakoram region; | We added (P3, L15-25, P2). |
| Improve the discussion of the results and in particular consider how to put your results into a broader context of w by others might be interested in your case study and the results; | To improve result and discussion part, we added a separate "Discussion" section where we discussed results in broader context (P29, L5 – P30, L30) |
| Highlight the limits about your method; | We highlighted limits of method (P30, L18-24) |
| Look again at the balance of the paper, which is a little unbalanced as there is a large section related to study and geologic settings, particularly compared to a shorter discussion part; | We readjusted paper and moved outcome of spatial analysis of controlling factors and landslide inventory analysis to "Result" section (P19, L1 – P23, L20). |
| Some figures are unnecessary (eg. fig.5, fig.7, fig.8), while others need to be improved (eg. fig. 3, the rainfall looks distributed within mountains as a meandering strip); | We deleted (Fig. 5, 7 and 8) as recommended. Rainfall pattern follows topography and this meandering is due to it.
However, we improved as advised! |
| Look at the replies provided during the first stage of the review, it seems that you did not check all the points raised (eg. anonymous Referee #3); | As per advice, I checked again. You are right; we actually did not change in response to two comments. Now, rephrased abstract (P1, L10-25) and discuss model used (in "Discussion"). |
| **Anonymous Referee #5** | |
| I do not agree with the classification of landslides used to build the inventory (Table 1). Why did you classify Debris flows and Rock falls under Shallow landslides category? They are different processes and urge to be separated. Moreover, in the second column what do you mean with Falls (Rock, Debris)? There is no scheme defining "debris falls"… | We separated as advised. (Page 19, L15) |
| A proper discussion comparing your methodology and related outcomes with other works on landslide susceptibility is totally missing and should be introduced. | We inserted as advised. (Page 29, L5 – P30, L23). |

| | |
|---|---|
| L. 19, pag. 1: in the abstract, there is no need to specify the commercial GIS software used in the study. Please replace "using ArcGIS 10.3" with "in a GIS environment". | We corrected (Page 1, L15). |
| L. 13, pag. 2: in 2006-> (2006) and "angle"->slope | We corrected (Page 2, L8). |
| L. 14-16, page. 2: Bad English. This sentence should be rephrased (E.g., controls->expresses, "If it positive" the verb is missing, "Later has ability…" the subject is missing…) | We corrected (Page 2, L10). |
| L. 19, pag. 2: add "an" before easier | We corrected (Page 2, L16). |
| L. 17-18, pag. 10: Could you please provide here a general motivation for the choice of the parameters? | We mentioned (P29, L 7-10) |
| Caption Figure 6: "Spatial Analysis": These graphs are not properly representing a spatial analysis but frequency distribution histograms… | We changed as advised (P21, L1). |
| Figure 7: a north arrow is missing | We deleted this as advised by Editor. |
| L. 20, pag. 13: by->to | We changed (P23, L 20). |
| Figure 9: I suggest using Field Observation in place of Field Reconnaissance here and in the related chapter. | We changed (P15, L1). |
| **Anonymous Referee #4** | |
| The abstract seems copy paste of some sentences forms the main body of the manuscript. Not need to describe how Himalaya was formed in the manuscript. Highlight what is existing in instability problem and what is your contribution | We rephrased the abstract. We deleted as per advice. |
| The introduction provides somehow sufficient background information, however, the authors need to provide an explanation about the necessity using the proposed models within the research in the particular area. | We have written motivation to use this model (P3, L1-15) |
| "Some geoscientist incorporate………" you have to put more consistent references. | We insert references (P2, L31) |
| In the introduction, the objectives and hypothesis you be established and findings shall be written in your result section. How your research is different from others and | We compared in discussion (P29, L18 – P30, L9) |

| | |
|---|---|
| what is your specific idea? should be mentioned. | |
| I am confused with your study area, is it just a buffer area of the road, right? But after reading this section and geology part, it seems more about regional scale containing unnecessary areas. | Now, we mentioned (P4, L10). |
| The geological description is out of the scope, it's like a report. You should describe the effect of geological units and structure and existence of landslide | Initially we gave an overview of geology and then discuss its effect in spatial distribution analysis (P20, L3-7) |
| How and why did you select 10 conditioning factors? What is the basis of that? | We mentioned (P29, L7). |
| It is not clear how selected conditioning factors influence generation of landslides? | We described in "Causative factors and spatial distribution analysis" (P19, L17 – P23, L20) |
| In this large area, only major faults play an important role? Included lineament attributes. | Slope angle and fault both play an important role. We included mapped regional and local faults but not the lineaments. |
| How did you make classes for continuous data such as slope, elevation, drainage proximity, curvature etc.? Do have you try to make a sensitivity analysis of different classes? If you aggregate the three first class, the final probabilities increase in these locations? You do not make sensitivity analysis on the different data and classes, but it is the first step to have a robust strategy of LSA and robust final susceptibility map! | We tried different classes within a parameter during spatial analysis and then used for production of landslide susceptibility map. |
| How did you prepare the rainfall map? Is that annual max rainfall, mean rainfall, event base rainfall or total rainfall etc? | We interpolated rainfall data of six weather stations to generate raster rainfall map. |
| Is the proximity of streams only one hydrological factor? You have DEM, why did not you generate others like TWI, SPI, STI etc? | It is a great idea and we shall include in our upcoming publication. As per scope of this paper, we did not included this. |
| Explain how changes in landcover control the spatial distribution of landslide in your study area. Did you compare imageries? | We explained (P23, L16-20) |
| A poor architect was presented in the methodological part. | We tried to give an overview of the methodology and data sets used. |
| What is the real improvement of the field reconnaissance survey? How many landslides have been corrected in term of | It is a nice idea and we shall include in our next publication. |

| | |
|---|---|
| locations and type? You have to put more details about the approach, results and improvements. You can take example about that in the parer of Fressard et al., 2013. They compared different inventories and the improvements of field survey. Please explain why you focus on thes landslide types... | |
| Give reference to SCP. It is not clear the accuracy assessment of landcover map. This should be in result part not in the methodological section. | We gave reference (Page 23, L 6-7). As per advice, we insert it in result part. |
| With threshold do you used to split the weighted map into four categories? How do you choose these values? This is an important point that is not justified or described in the text. | We mentioned (P23, L23-25). |

[revised manuscript text omitted]

---

## Author Response (AR3)

First, we would like to thank you for evaluation and highlighting the deficiencies in the manuscript. It is indeed valuable addition and help us to improve our manuscript. It looks far better than what I submitted initially. Thank you very much for your valuable comments.

Please find below our response to comments.

| Comments | Response |
|---|---|
| scale of the figures: use "km" and not "Kilometers"; also please keep the same format (and style), therefore, in few figures, delete the word "scale" that it is not necessary; | We changed as suggested. |
| (a)(b)(c)... letters of the figure: also in this case use the same style and font in all the figures; | We changed as suggested. |
| conclusions: avoid to cite the name of the software or the number of figures, since they are not necessary for the conclusions; | We removed as advised. |
| I would recommend again a general check of English. | We checked the language (grammar) and typing mistakes. |